# Gain, not concomitant changes in spatial receptive field properties, improves task performance in a neural network attention model

**Kai J Fox\*†, Daniel Birman\*†‡, Justin L Gardner**

Department of Psychology, Stanford University, Stanford, United States

**Abstract** Attention allows us to focus sensory processing on behaviorally relevant aspects of the visual world. One potential mechanism of attention is a change in the gain of sensory responses. However, changing gain at early stages could have multiple downstream consequences for visual processing. Which, if any, of these effects can account for the benefits of attention for detection and discrimination? Using a model of primate visual cortex we document how a Gaussian-shaped gain modulation results in changes to spatial tuning properties. Forcing the model to use only these changes failed to produce any benefit in task performance. Instead, we found that gain alone was both necessary and sufficient to explain category detection and discrimination during attention. Our results show how gain can give rise to changes in receptive fields which are not necessary for enhancing task performance.

**\*For correspondence:**
kaifox@stanford.edu (KJF);
dbirman@uw.edu (DB)

†These authors contributed equally to this work

**Present address:** ‡Department of Biological Structure, University of Washington, Washington, United States

**Competing interest:** The authors declare that no competing interests exist.

## Editor's evaluation

This manuscript combines human behavioral experiments using a categorization task and a convolutional neural network model to test different mechanisms that may support attention-related improvements in perception. Through carefully controlled manipulations of computational architecture and parameters, the authors dissociate the effects of tuning gain vs. tuning shifts. They conclude that increases in gain are the primary means by which attention improves behavioral performance.

## Introduction

Deploying goal-directed spatial attention towards visual locations allows observers to detect targets with higher accuracy (*Hawkins et al., 1990*), faster reaction times (*Posner, 1980*), and higher sensitivity (*Sagi and Julesz, 1986*) providing humans and non-human primates with a mechanism to select and prioritize spatial visual information (*Carrasco, 2011*). These enhanced behavioral responses are accompanied by both an increase in the gain of sensory responses near attended locations (*Connor et al., 1996*; *McAdams and Maunsell, 1999*) and changes in the shape and size of receptive fields, typically shrinking and shifting towards the target of attention (*Ben Hamed et al., 2002*; *Womelsdorf et al., 2006*; *Anton-Erxleben et al., 2009*; *Klein et al., 2014*; *Kay et al., 2015*; *Vo et al., 2017*; *van Es et al., 2018*). These changes in neural representation are thought to contribute to behavioral enhancement, but because both gain and changes in spatial properties co-occur in biological systems, it is not possible to disentangle them. Computational models of the visual system allow us to design experiments to independently examine the effects of such changes (*Lindsay and Miller, 2018*; *Eckstein et al., 2000*).

Shrinkage and shift of receptive fields toward attended targets has been observed in both single unit (*Womelsdorf et al., 2006*; *Anton-Erxleben et al., 2009*) and population (*Klein et al., 2014*; *Vo et al., 2017*; *Fischer and Whitney, 2009*; *van Es et al., 2018*) activity, and has been suggested to lead to behavioral enhancement through a variety of possible mechanisms (*Anton-Erxleben and Carrasco, 2013*). For example, receptive field changes might magnify the cortical representation of attended regions (*Moran and Desimone, 1985*), select for relevant information (*Anton-Erxleben et al., 2009*; *Sprague and Serences, 2013*), reduce uncertainty about spatial position (*Vo et al., 2017*), increase spatial discriminability (*Kay et al., 2015*; *Fischer and Whitney, 2009*), or change estimates of perceptual size (*Anton-Erxleben et al., 2007*). Compression of visual space is also observed just prior to saccades and thought to shift receptive fields towards the saccade location (*Zirnsak et al., 2014*; *Colby and Goldberg, 1999*; *Merriam et al., 2007*) and maintain a stable representation of visual space (*Kusunoki and Goldberg, 2003*; *Tolias et al., 2001*; *Ross et al., 1997*; *Duhamel et al., 1992*).

Shrinkage and shift of receptive fields has also been hypothesized to occur as a side effect of increasing gain of neural responses (*Klein et al., 2014*; *Compte and Wang, 2006*), thus raising the question of which of these physiological effects could be responsible for enhanced perception. When gain is asymmetric across a receptive field, the overall effect will be to shift the receptive field location towards the side with the largest gain. Similarly, asymmetric gain can be expected to change spatial tuning properties such as the size and structure of the receptive field. These concomitant changes of receptive field size, location, and structure could improve perceptual performance through the mechanisms described above, or could be an epiphenomenological consequence of increasing gain. Increasing gain by itself has also been hypothesized to be a mechanism for improving perceptual performance, because response gain can increase the signal-to-noise ratio and make responses to different stimuli more discriminable (*McAdams and Maunsell, 1999*; *Cohen and Newsome, 2008*). Moreover, larger responses for attended stimuli can act to select relevant information when read-out through winner-take-all mechanisms (*Lee et al., 1999*; *Pelli, 1985*; *Pestilli et al., 2011*; *Palmer et al., 2000*; *Hara et al., 2014*).

We took a modeling approach to ask what effects gain changes incur on spatial receptive field structure when introduced at the earliest stage of visual processing and to ask which effects would improve behavioral performance. We modified a convolutional neural network (CNN) trained on ImageNet categorization to test various hypotheses by implementing them as elements of the model architecture. CNN architectures can be designed to closely mimic the primate visual hierarchy (*Yamins et al., 2014*; *Kubilius et al., 2018*). Training 'units' in these networks to categorize images leads to visual filters that show a striking qualitative resemblance to the filters observed in early visual cortex (*Krizhevsky et al., 2012*) and the pattern of activity of these units when presented with natural images is sufficient to capture a large portion of the variance in neural activity in the retina (*McIntosh et al., 2016*), in early visual cortex (*Cadena et al., 2019*), and in later areas (*Güçlü and van Gerven, 2015*; *Cichy et al., 2016*; *Eickenberg et al., 2017*; *Khaligh-Razavi and Kriegeskorte, 2014*; *Yamins et al., 2014*). Cortical responses and neural network activity also share a correlation structure across natural image categories (*Storrs et al., 2020*). These properties of CNNs make them a useful tool which we can use to indirectly study visual cortex, probing activity and behavior in ways that are impractical in humans and non-human primates (*Lindsay and Miller, 2018*).

Using simulations based on a CNN observer model we found that gain changes introduced at the earliest stage in visual processing improved task performance with a magnitude comparable to that measured in human subjects. While these gain changes also induced changes in receptive field location, size, and spatial structure similar to that reported in physiological measurements, these changes were neither necessary nor sufficient for improving model task performance. More specifically, we designed a simple cued object-detection task and measured improved human performance on trials with focal attention. Using CORnet-Z (*Kubilius et al., 2018*), a CNN whose architecture was designed to maximize similarity with the primate visual stream, we measured a similar improvement in detection performance when a Gaussian gain augmented inputs coming from a 'cued' location. We found that the network mirrored the physiology of human and non-human primates: units shifted their center-of-mass toward the locus of attention and shrank in size, all in a gain-dependent manner. We isolated each of these physiological changes to determine which, if any, could account for the benefits to performance. A model with only gain reproduced the benefits of cued attention while models

with only receptive field shifts, shrinkage, or only changes in receptive field structure were unable to provide any benefit to task performance. These results held for both an object detection task and a category discrimination task. Gain applied or removed at the last stage of processing in the CNN observer model demonstrated that gain was both necessary and sufficient to account for the benefits in task performance of the model.

## Results

We characterized the ability of human observers to detect objects in a grid of four images, with or without prior information about the object's possible location (**Figure 1**). Observers were given a written category label, for example 'ferris wheel', and shown five exemplar images of that category (Category intro, **Figure 1a**). This was followed by a block of 80 trials in which observers tried to detect the presence or absence of the target category among the four images in the grid (Each trial, **Figure 1a**). Half of the 80 trials had focal cues and 50% of the focal (and distributed) trials included a target image. On focal trials, a cue indicated with 100% validity the grid quadrant that could contain a target while on distributed trials no information was given as to where an image of the target category could appear. Distractor images were randomly sampled from the nineteen non-target image categories. Stimulus durations were sampled uniformly from 1 (8.3 ms), 2 (16.7), 4 (33.3), 8 (66.7), 16 (133.3), or 32 (267.7) frames (Stimulus, 8.3ms per frame, **Figure 1a**). Image grids were masked before and after stimulus presentation by shuffling the pixel locations in the stimulus images, ensuring that the luminance during each trial remained constant. Observers had 2 s to make a response and each trial was followed by a 0.25 s inter-trial interval. Observers completed one training block on an unused category prior to data collection.

Human observers improved their performance on this detection task when given a focal cue indicating the potential location of a target (**Figure 1b**). We quantified human performance by computing sensitivity, d', as a function of stimulus duration separately for focal and distributed conditions. Across all observers the $d'$ function was best fit as:

$$d'(ms) = \alpha \log(163.6ms + 1) \tag{1}$$

where $\alpha$ scaled the function for the focal condition. At a stimulus duration of 8.3ms (one frame) observers were near chance performance regardless of cueing condition. On distributed trials observers exceeded threshold performance ($d' = 1$) at a stimulus duration of 155ms, 95% CI [135, 197]. For focal trials, the same threshold was reached with only a 38ms [32, 43] stimulus duration, demonstrating a substantial performance benefit of the focal cue. We found that $d'$ in the focal condition was higher than in the distributed condition, average increase across observers $\alpha = 1.67\times$ [1.57, 1.74].

Using a drift diffusion model, we found that the majority of this performance benefit came from improved perceptual sensitivity, rather than speed-accuracy trade off. We assessed this by fitting a drift diffusion model to the reaction time and choice data (**Wagenmakers et al., 2007**). Drift diffusion models assume that responses are generated by a diffusion process in which evidence accumulates over time toward a bound. We used the equations in **Wagenmakers et al., 2007** to transform each observer's percent correct, mean reaction time, and reaction time variance for the twenty categories and two focal conditions into drift rate, bound separation, and non-decision time. The drift rate parameter is designed to isolate the effect of external input, the non-decision time reflects an internal delay before stimulus processing begins, and the bound separation is a proxy for how conservative observers are. Comparing the drift rate parameter we observed a similar effect to what was described above for $d'$: the average drift rate across observers in the focal condition was 1.61×, 95% CI [1.39, 1.77] the drift rate in the distributed condition. This suggests that the majority of the performance gain observed in the $d'$ parameter came from increased stimulus information. We did find that the other parameters of the drift diffusion model were also sensitive to duration and condition, but in opposite directions. We found larger bound separation at longer stimulus durations and on focal trials (focal bound-separation 1.57× distributed [1.37, 1.75]), consistent with observers being more conservative on trials where more information was available. But this increase in cautiousness was offset by a shorter non-decision time on focal trials (0.26 s) compared to distributed (0.38, [0.34, 0.41]).

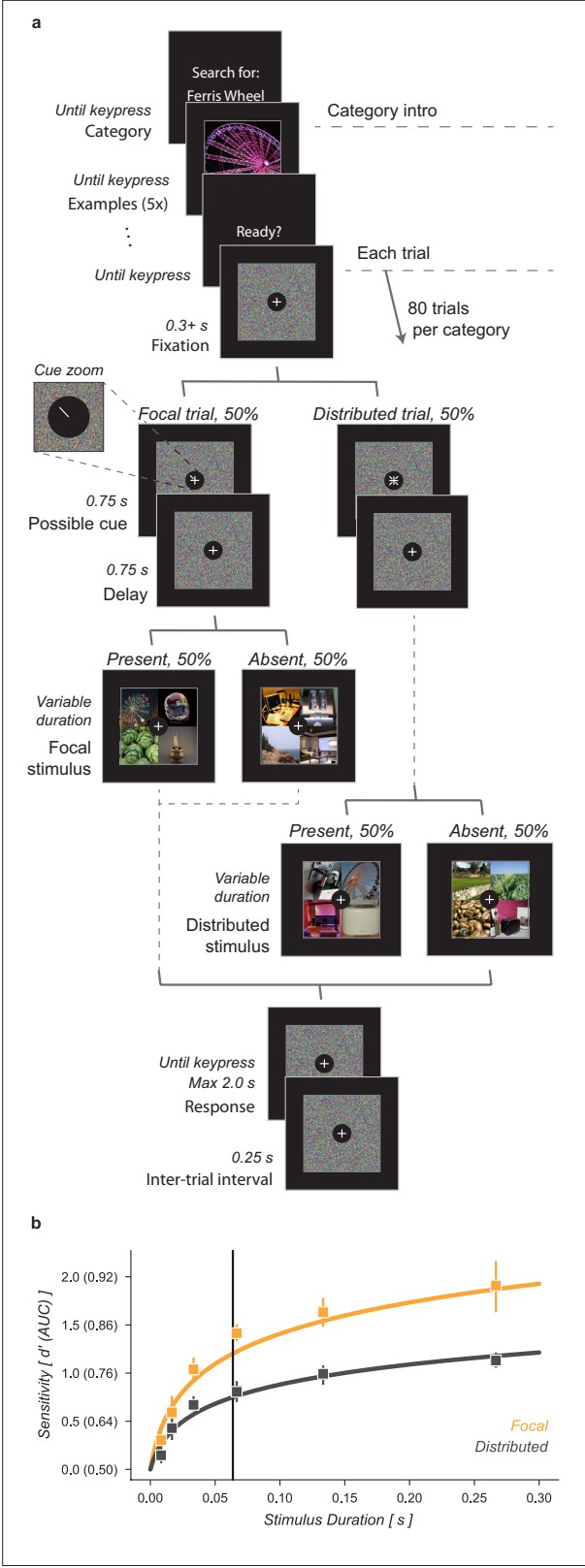

**Figure 1.** Cued object detection task. (**a**) Observers were asked to perform object detection with or without a spatial cue. At the start of a block, observers were shown five examples of the target category. This was followed by 80 trials: 40 with a spatial cue indicating the possible target quadrant and 40 with no prior information. Stimulus presentation was pre and post-masked. The stimuli consisted of a composite grid of four individual object

*Figure 1 continued on next page*

*Figure 1 continued*

exemplars. The target category was present in 50% of trials and always in the cued location on focal trials. Human observers used a keyboard to make a fast button response to indicate the target presence before moving on to the next trial. (**b**) Human observers showed a substantial improvement in performance when given a focal cue indicating the quadrant at which the target might appear. Vertical line at 64ms indicates the duration at which the best-fit $d'$ curve for the Distributed condition matched the CNN observer model performance without gain. Markers indicate the median and error bars the 95% confidence intervals (n=7 observers).

Having shown that a spatial cue provides human observers with increased stimulus information in this task, we next sought to show that a neural network model of the human visual stream could replicate this behavior under similar conditions. We used a convolutional neural network (CNN) model, CORnet-Z (**Kubilius et al., 2018**), a neural network designed to mimic primate V1, V2, V4, and IT

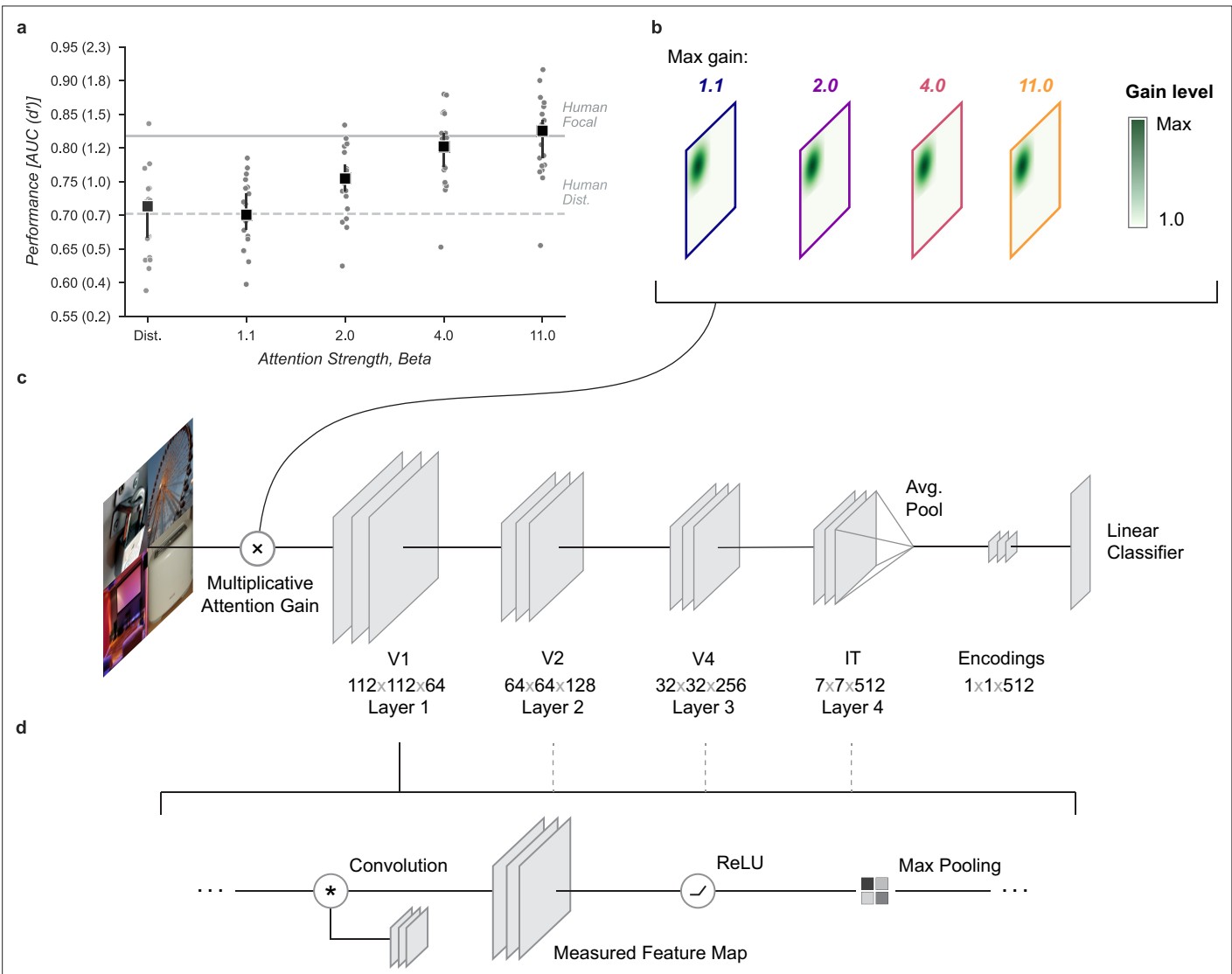

**Figure 2.** Neural network observer model. (**a**) Using a Gaussian gain the neural network observer was able to replicate the benefit of spatial attention for human observers. Human performance is shown at a stimulus duration of 64ms which provided the closest match to the convolutional neural network (CNN) performance without gain. Black markers indicate the median by category and error bars the 95% confidence intervals (n=20 categories). (**b**) The Gaussian gain was implemented by varying the maximum strength of a multiplicative gain map applied to the 'cued' quadrant. (**c**) The gain was applied prior to the first layer of the CNN. The neural network observer model consisted of a four layer CNN with linear classifiers applied to the output layer. Individual classifiers were trained on examples of each object category. (**d**) Each of the four convolutional layers consisted of a convolution operation, a rectified linear unit, and max pooling. Unit activations were measured at the output of each layer.

and optimized to perform object recognition for images at a similar scale to our task. CORnet-Z is a four-layer CNN with repeated convolutional, rectified linear units (ReLU), and pooling (*Figure 2d*). We used pretrained weights which were optimized for object categorization on ImageNet (*Deng et al., 2009*). To perform our category detection task, we added a fully connected output layer for each category and trained the weights of that layer to predict the presence of the twenty object categories selected for this study, thus creating a neural network observer model, that is a model designed to idealize the computations human observers perform in the four-quadrant object detection task. We applied the observer model to a task analogous to the one human observers performed (*Figure 2c*). The prediction layers added to the end of the model provided independent readouts for the presence or absence of the different target categories (Linear classifier, *Figure 2c*). These output layers were trained on a held out set of full-size images from each category. On a separate held out validation set of 100 images, the trained prediction layers achieved a median AUC of 0.90, range [0.77, 0.96].

To examine the computational mechanisms that could underlie the performance benefit of focal cues we added a multiplicative Gaussian gain centered at the location of the cued image (*Figure 2b*, Gaussian width 56 px). We applied this gain at the first layer of the model and tested various strengths of gain.

To align the human and model performance we took the performance of the model in the distributed condition (Distributed, *Figure 2a*) and found the stimulus duration at which observers in the distributed condition of the human data matched this performance level (64ms, *Figure 1b*). We then scaled up the amplitude of the Gaussian gain incrementally and found that we could mimic the performance enhancement of human spatial attention by setting the maximum of the Gaussian gain field to approximately $4\times$. The model with this level of gain had a median AUC across categories of 0.80, 95% CI [0.77, 0.82] compared to 0.71 [0.67, 0.72] without gain and a median AUC improvement of 0.09 [0.08, 0.12] within each category.

The gain strengths necessary to induce an increase in task performance in the neural network observer model were relatively large compared to the gain due to directed attention observed in measurements of single unit (*Luck et al., 1997*; *Treue and Martínez Trujillo, 1999*) and population (*Birman and Gardner, 2019*) activity. We attribute this difference to the lack of any non-linear "winner-take-all" type of activation in the CNN. In the primate visual system, it is thought that non-linearities such as exponentiation and normalization can accentuate response differences (*Reynolds and Heeger, 2009*; *Carandini and Heeger, 2012*) and act as a selection mechanism for sensory signals (*Pestilli et al., 2011*). We tested whether similar non-linear mechanisms would allow for smaller gain strengths to be amplified to the range needed by our model. This was tested by raising the activations of units to an exponent before re-normalizing the activation of all units at the output of each layer (see Materials and methods for details). This has the effect of amplifying active units and further suppressing inactive ones. Using this approach, we found that a relatively small gain of $1.1\times$ combined with an exponent of 3.8 led to a significantly larger effective gain of $1.37\times$ after just one layer (*Figure 3j*). This form of non-linearity is consistent with the finding that static output non-linearities in single units range from about 2–4 (*Gardner et al., 1999*; *Albrecht and Hamilton, 1982*; *Sclar et al., 1990*; *Heeger, 1992*) and suggests a plausible physiological mechanism by which the larger gains predicted by our model could be implemented. Repeated use of exponentiation and normalization in successive layers of the visual system could produce an even larger effective gain. To avoid training a new convolutional neural network (CNN) and possibly violate the close relationship between the primate visual system and the CNN we studied, we continued our analysis without introducing an exponentiation and normalization step.

The Gaussian gain could have its effect on the neural network observer model's performance by increasing the activation strength of units with receptive fields near the locus of attention. These changes in activation strength might directly modify behavior, or work indirectly through mechanisms such as changes in receptive field size, location, or spatial tuning. We observed all of these effects in our model (*Figure 3*). To measure receptive fields we computed the derivative of each unit with respect to the input image and then fit these with a 2D Gaussian (see Materials and methods for details). We found that the gain caused receptive fields to shift and shrink toward the locus of attention (*Figure 3a and b*). The information provided by individual units in the model also changed, increasing for units on the border of the cued quadrant (*Figure 3c*). The receptive field shift and shrinkage were magnified in deeper layers of the model (*Figure 3d and e*) consistent with physiological observations (*Klein*

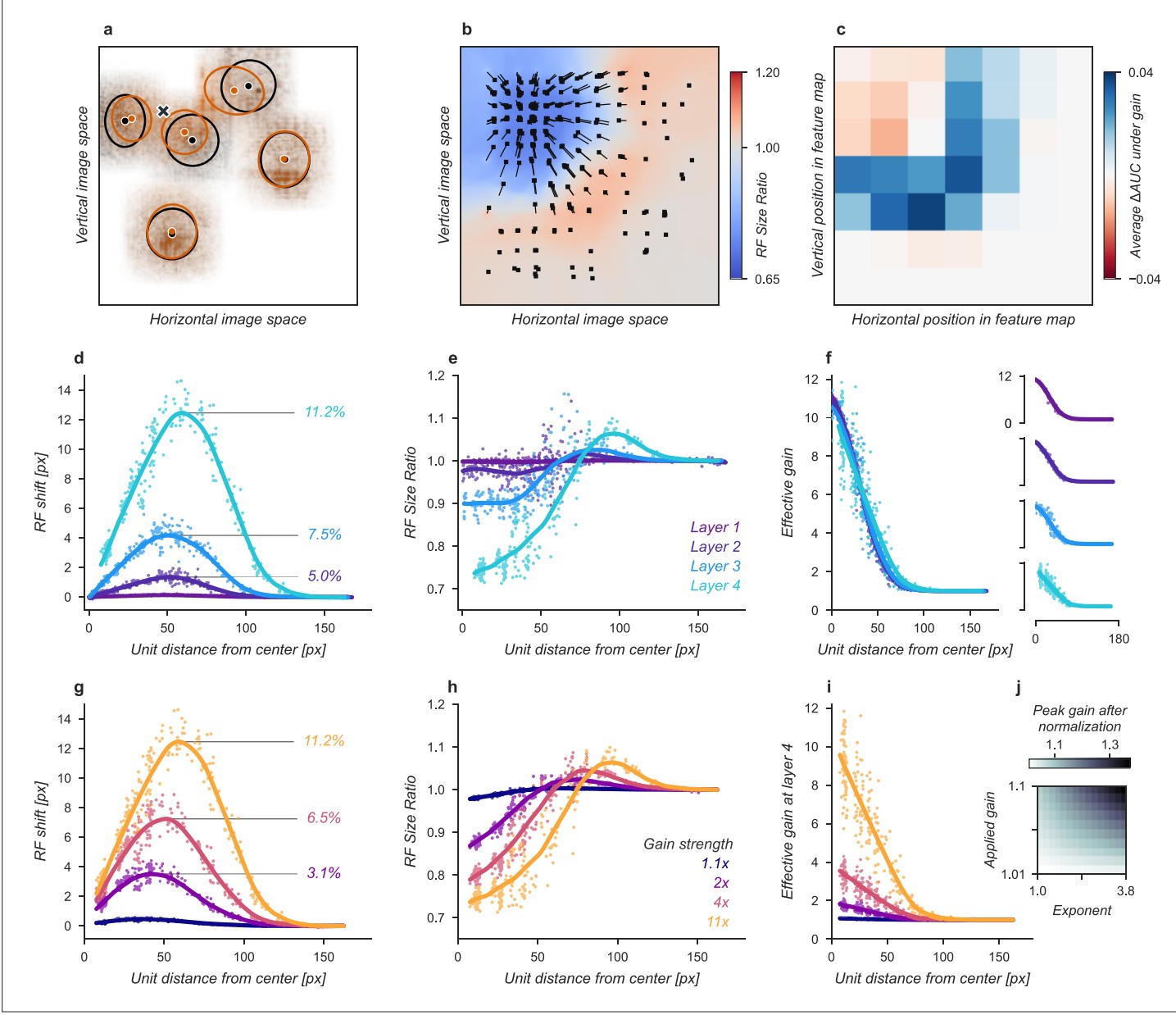

**Figure 3.** Effects of Gaussian gain on neural network units. (**a**) The Gaussian gain applied to Layer 1 units caused the measured receptive field (RF) of units in Layer 4 to shift (black ellipse, original; brown ellipse, with gain) toward the locus of attention (black ×). (**b**) A 2D spatial map demonstrates the effects of Gaussian gain in Layer 4: shift of RF center position (black arrows), shrinking RF size near the attended locus (blue colors) and an expansion of size near the gain boundaries (red colors). (**c**) 7 × 7 map of the output layer before averaging, showing the change in AUC caused by the addition of Gaussian gain. Each pixel's ΔAUC is computed by projecting the activations at that location for composite grids with target present and absent on the decision axis and then calculating the difference in AUC between a model with and without Gaussian gain. The map demonstrates that units overlapping the borders of the composite grid have the largest change in information content when Gaussian gain is applied. (**d,e**) Scatter plots demonstrate that each layer magnifies the effect of the gain on RF shift and RF size. The RF shift percentages are the ratio of pixel shift at the peak of the curve relative to the average receptive field size, measured as the full-width at half-maximum. (**f**) Later layers do not magnify the effective gain (shown for an 11× gain), which stays constant across layers. (**g**) Gain strength influences the size of RF position shifts, RF size (**h**), and effective gain (**i**). (**j**) Adding an additional non-linear normalizing exponent at the output of each layer allows for much smaller gains to be magnified across layers. Markers in all panels indicate individual sampled units from the model. Lines show the LOESS fit for visualization.

*et al., 2014*). The gain in activation strength propagated through the network without modification (*Figure 3f*). To measure the effective gain experienced by the layer four units (*Figure 3i*), we computed the ratio of the standard deviations of unit activations at the output of each layer (*Figure 2d*) with and without gain applied. All three observed effects: receptive field shift, shrinkage and expansion, and effective gain were directly related to the gain strength at the input layer (*Figure 3g–i*). All of these changes have been proposed as mechanisms that could account for the behavioral benefits of attention (*Anton-Erxleben and Carrasco, 2013*; *Moran and Desimone, 1985*; *Anton-Erxleben et al., 2009*; *Sprague and Serences, 2013*; *Vo et al., 2017*; *Kay et al., 2015*; *Fischer and Whitney, 2009*; *Anton-Erxleben et al., 2007*). We designed models to try to isolate these effects with the goal of testing their independent contributions to task performance.

We next sought to test whether receptive field shifts alone could account for the behavioral benefits of the neural network observer model. To do this, we built a model variant that could shift receptive fields without introducing gain. To develop an intuition for how this could affect perceptual reports, consider a CNN with just four units in a 2 × 2 grid with each unit having its receptive field centered on one image in the composite. When shown a composite grid of four images, a logistic regression using the output of these four units would receive one quarter the information it expects from being trained on full size images. Shifting the receptive fields of the three non-target units to overlap more with the cued image could add additional task-relevant information to the output, much as was observed for units with receptive fields overlapping multiple images in the Gaussian gain attention model (*Figure 3c*).

We designed a variant of our model that could be used to test the hypothesis that receptive field shifts alone are responsible for the behavioral enhancement (*Figure 4*). In this model, we re-wired the units in the first layer to reproduce the effect of Gaussian gain. The re-wiring was designed so that receptive fields in the fourth layer matched their shift with the Gaussian gain model (*Figure 3g*). To mimic those shifts, we changed the connections between the input image pixels and layer one (*Figure 4a*). This manipulation worked as designed and changed the receptive field locations and size (*Figure 4b–d*) but since no gain was added to the model, the overall responsiveness of units remained constant (*Figure 4e*). Because receptive field shifts due to gain are not the result of actual rewiring it is unsurprising that the shift and shrinkage in this model variant are only qualitatively matched to those caused by the original Gaussian gain. Note that the effective gain of individual units in layer four *did* change for individual images, a result of each unit receiving different inputs, but the average change across images was zero.

We found that the model with receptive field shifts but no gain had no effect on task performance, demonstrating that receptive field shifts are not key for the improvement in task performance observed with Gaussian gain (*Figure 4f*). The model imitating shifts from 4× Gaussian gain had a median AUC across categories of 0.71, 95% CI [0.66, 0.73] compared to 0.71 [0.67, 0.72] in the baseline model, median change in AUC of –0.01 [-0.02, 0.01] within each category.

Another way to understand the possible effect of the Gaussian gain on task performance is to note that the spatial tuning profile of units is 'shifted' towards the locus of attention: sensitivity is enhanced closer to the locus of attention, but the receptive field itself has not truly moved in the manner studied by the previous model. If different parts of a receptive field receive asymmetric gain, as expected for Gaussian gain then the local structure of the receptive field has been changed (*Figure 5a*). We designed another model variant to test the hypothesis that these local changes in receptive field structure might be sufficient to explain changes in task performance without inducing receptive field shifts or gain. To implement this model at layer $L$, we examined the effect of the Gaussian gain on each unit (green differential gain, *Figure 5a*). We normalized this differential gain within each unit's receptive field to prevent any overall gain effect and re-scaled the unit's kernel accordingly. Overall, this manipulation of each unit's kernel preserved a portion of the receptive field shift effect in a gain-dependent manner but guaranteed that there was no effective gain.

The receptive field structure model was designed to only change the spatial tuning of individual units without inducing gain, which naturally caused some shifts in the measured receptive field size and location (solid lines and markers, *Figure 5b–d*) but these were smaller than the effects observed under Gaussian gain (dashed lines). The normalization prevented the model from introducing any spatial pattern of gain change (*Figure 5e*). Note that there were still small changes in overall sensitivity of units in this model, for example, the 4× model had an average gain of 1.08, 95% CI [1.07,

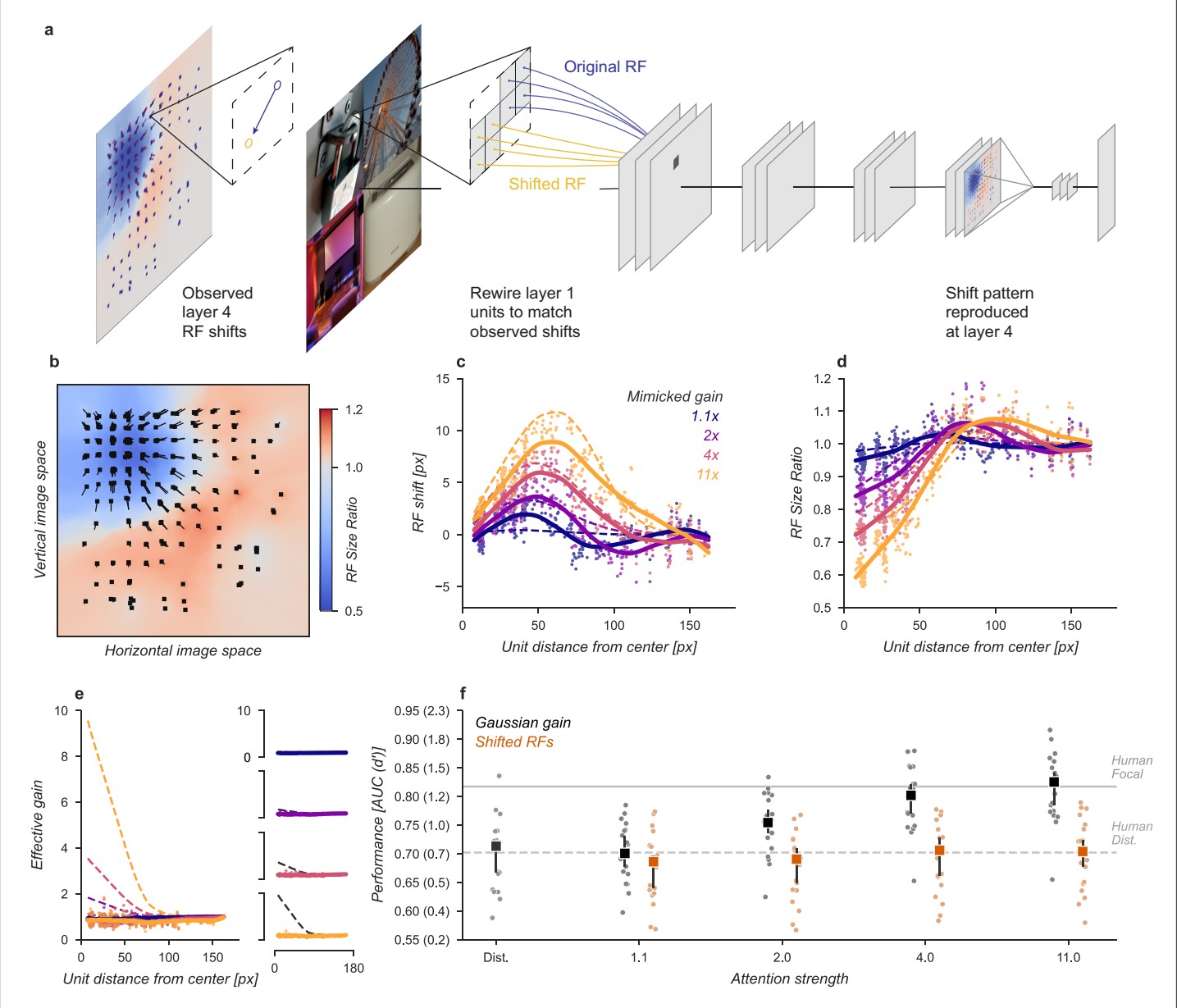

**Figure 4.** Receptive field shift model. (**a**) To mimic the effects of the Gaussian gain on receptive field position without inducing gain in the model we re-assigned the inputs to units in Layer 1. This re-assignment was performed so that the pattern of receptive field shift in Layer 4 would match what was observed when the Gaussian gain was applied. (**b**) The observed pattern of receptive field shifts and shrinkage is shown for a sample of units in layer 4, qualitatively matching the effects of the Gaussian gain. (**c**) RF shift is shown for sampled units (markers) and the LOESS fit (solid lines) compared to the effect in the Gaussian gain model (dotted lines). (**d**) Conventions as in c for the RF size change. (**e**) Conventions as in (**c,d**) for the effective gain of units. (**f**) The behavioral effect of shifting receptive fields is shown to be null on average across categories when compared to the effect of Gaussian gain. Large markers indicate the median performance, small markers the individual categories (n=20), and error bars the 95% confidence intervals.

1.09] across all units, which we attribute to the fact that inputs to a unit may exhibit correlations due to spatial structure. These receptive field changes and small gain effects were distinct from those observed under Gaussian gain (*Figure 5c–e*).

The receptive field structure model, like the shift model, was unable to account for the behavioral effects of the Gaussian gain. No matter where in the model we changed the receptive field structure, and even when applied at all layers, the average performance across categories remained flat (*Figure 5f*). Compared to the median distributed AUC across categories of 0.71 [0.67, 0.72], the sensitivity model applied to all layers had a median AUC across categories of 0.69 [0.65, 0.72] when

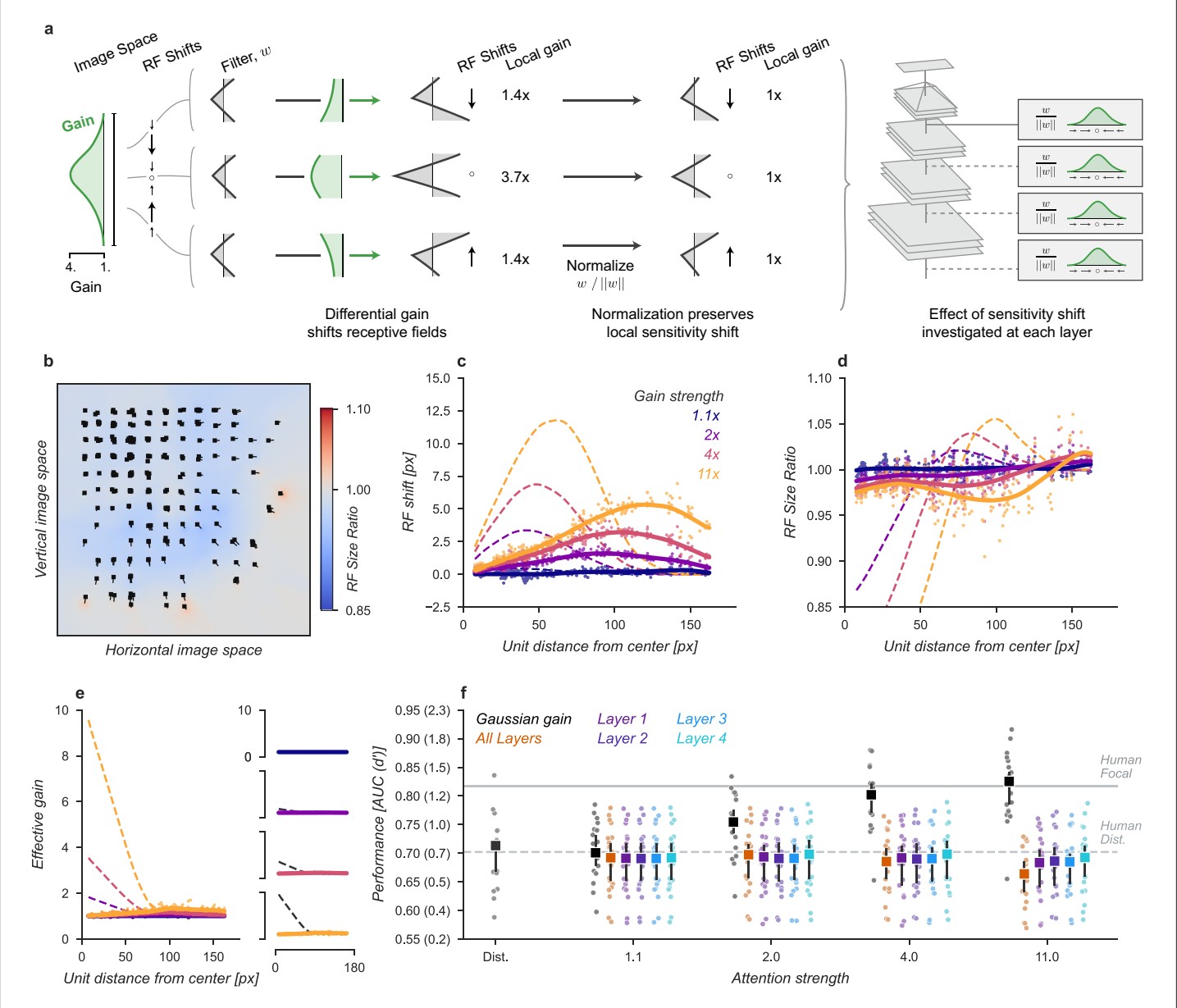

**Figure 5.** Receptive field structure model. (**a**) We adjusted the kernels of each convolutional neural network (CNN) unit according to the effect of a Gaussian gain, subtly shifting the the sensitivity within individual units. To avoid inducing a gain change, we then normalized each units output such that the sum-of-squares of the weights was held constant, ensuring the local gain at that unit remained at 1×. This model was implemented individually at each layer, replicating the effect of a Gaussian gain of 1.1×–11× as well as at all layers at once. (**b–f**) conventions as in *Figure 4*.

imitating gain of $1.1\times$, 0.70 [0.65, 0.72] for $2\times$ gain, 0.69 [0.65, 0.71] for $4\times$ and 0.66 [0.63, 0.69] for $11\times$. Each of these conditions resulted in a median AUC change within category of –0.02 [-0.03, 0.00], –0.01 [-0.03, 0.00], -0.02 [-0.04,–0.01], and -0.04 [-0.05,–0.03], respectively. When applied to early layers, we observed a slight drop in performance, which we attribute to how this model directly alters the kernels in the CNN. These changes break the assumption that the CNN kernels at each layer are consistent with those that were optimized when the model weights were trained.

The Gaussian gain also caused units to shrink and expand their receptive fields across the visual field (*Figure 3b*). These changes might modify the information content received at the output layer, improving or hurting performance. We designed a model variant to test the hypothesis that shrinkage and expansion of receptive fields, without shift or gain, might be sufficient to explain the behavioral effect (*Figure 6*). To implement this model, we took the observed change in receptive field size at

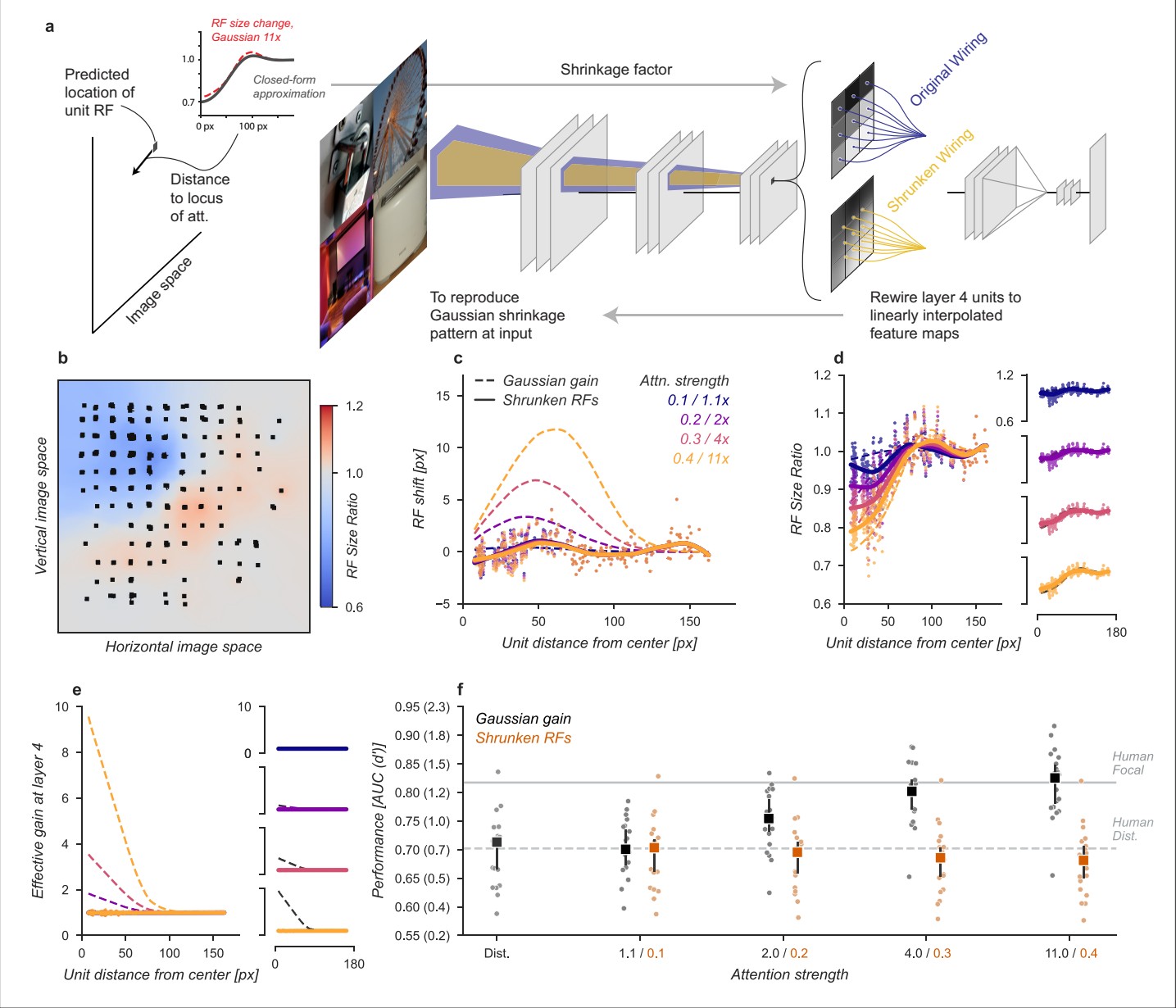

**Figure 6.** Shrinkage model. (**a**) To create shrinkage at layer 4 matched with the effects observed under Gaussian gain we re-assigned the connections between layers 3 and 4 according to a parameterized approximation of the shrinkage effect as a function of distance from the locus of attention. This re-scaling of connections changed the size of receptive fields without moving them in space or modifying their gain. (**b–f**) conventions as in previous figures.

layer 4 and then re-scaled the connections between layers three and four to mimic the observed effect. Because the kernels were scaled in space this manipulation has no effect on effective gain or receptive field position. Specifically, we approximated the shrinkage of the sampled units using a parameterized equation (*Equation 5*) that provides a shrinkage factor for every unit in the model (*Figure 6a*). We then re-wired the connections between layer three and four using linear interpolation to approximate the necessary change in scaling.

After re-wiring, units' receptive fields retained the same overall position, but were scaled to qualitatively match the observed effects under Gaussian gain (*Figure 6b–d*). The size changes do not match perfectly with those under Gaussian gain because we enforced symmetry in two ways: first, by parameterizing the shrinkage and expansion we enforced symmetry around the locus of attention, and second, because the observed receptive field changes were often asymmetric but we implemented a

symmetric linear scaling. These simplifications were necessary to reduce the complexity of implementation. By design, the shrinkage and expansion effects scaled by attention strength (*Figure 6d*) and induced neither gain-dependent shifts (*Figure 6c*) or effective gain (*Figure 6e*).

The shrinkage model was unable to account for improved task performance with Gaussian gain. The average performance across categories remained flat (*Figure 6f*). Compared to the median distributed AUC across categories of 0.71 [0.67, 0.72], the shrinkage model applied to all layers had a median AUC across categories of 0.70 [0.66, 0.72] when imitating gain of $1.1\times$, 0.70 [0.66, 0.71] for $2\times$ gain, 0.69 [0.65, 0.70] for $4\times$ and 0.68 [0.65, 0.70] for $11\times$. Each of these conditions resulted in a median AUC change within category of -0.01 [-0.01,–0.00], -0.01 [-0.01,–0.01], -0.02 [-0.02,–0.01], and –0.02 [–0.03, –0.01], respectively. We again observed drops in performance, which we attribute to how the kernels have been altered.

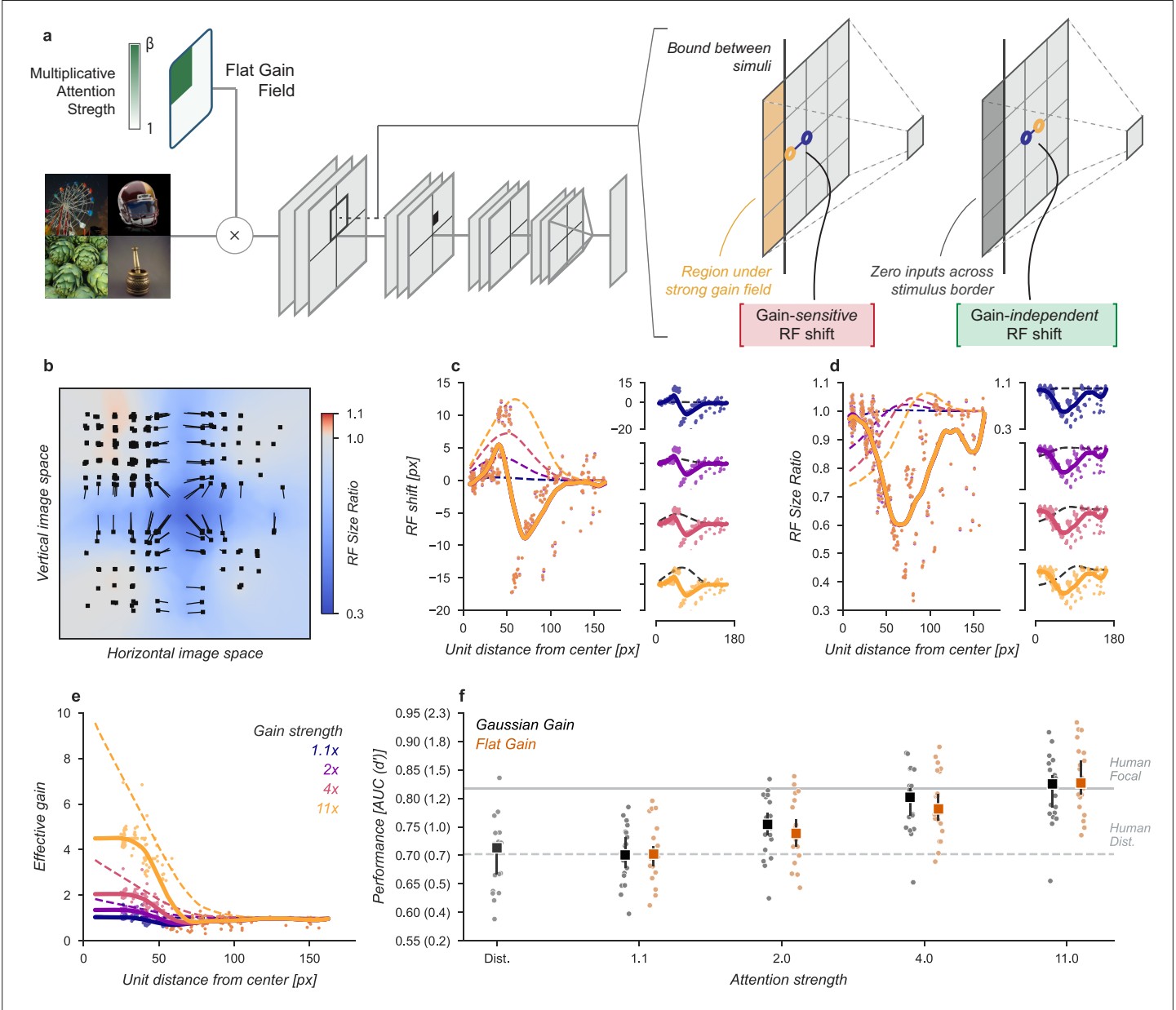

**Figure 7.** Gain-only model. (**a**) To create a gain effect without modifying the receptive fields of units, we applied a flattened gain field, with the gain set to the average of the original Gaussian gain for each attention strength. The flat gain alone causes units to shift their receptive field at the boundary between the four stimulus quadrants. To modify gain while ensuring shifts were gain-independent, we computed the four quadrants separately with zero padding and then concatenated the results. (**b–f**) Conventions as in previous figures.

Having ruled out that receptive field shift, shrinkage, or changes in spatial tuning could account for the improved task performance in our neural network observer, we next designed a model to amplify signals in the cued quadrant without these other effects and found that this model was able to explain the improved task performance observed with cued attention. In the original Gaussian gain model, an asymmetry in gain was introduced in the receptive fields of the units, causing size and location changes in the receptive fields. To remove this effect, we flattened the gain within the cued quadrant (*Figure 7a*) by setting the gain at each pixel to the average of the Gaussian gain across the entire quadrant. By itself, this change has the unintended consequence that units partly overlapping the cued quadrant will still shift in a gain-dependent manner. To remove this effect, we split the CNN feature maps into the four quadrants and computed these separately with padding and concatenated the results. This forces all units in the model to receive information about only a single quadrant. These manipulations did result in shifts in receptive field location and size for units at the borders (*Figure 7b–d*), but by design these were independent of the gain strength.

Using the gain-only model we were able to reproduce the improved task performance of the original Gaussian gain (*Figure 7*). The gain-only model induced the same pattern of receptive field shift and size change at all gain strengths (*Figure 7b–d*) and a flat effective gain within the cued quadrant (*Figure 7e*). We found that increasing the strength of a flat gain was sufficient to capture the full performance improvement of the original model (*Figure 7f*). The median AUC across categories of the $4\times$ flat gain model was 0.78, 95% CI [0.76, 0.83] compared to 0.80 [0.77, 0.82] for the $4\times$ Gaussian gain model. The confidence intervals in flat gain and Gaussian gain performance overlapped at all gain strengths, with a difference of 0.00 [-0.00, 0.02] at 1.1× gain, –0.01 [-0.02, 0.00] at 2× gain, –0.01 [-0.02, 0.00] at 4× gain, and 0.02 [0.00, 0.04] at 11× gain.

Having found that the improved task performance could be explained not by receptive field changes, but instead by the change in the overall gain, we asked whether gain propagated through the network was both necessary and sufficient to explain this effect. To test necessity and sufficiency, we ran the task images through the Gaussian gain model (first row, *Figure 8a*) and measured the effective gain propagated to units in the final layer output (7 × 7 × 512, before averaging). We averaged these effective gains over features to obtain a propagated gain map (Layer 4 feature map, 7 × 7, *Figure 8b*). To test the hypothesis that this propagated gain was sufficient to account for improved performance in the task, we re-applied it to the output layer of a model with no gain applied to the inputs.

We found that the propagated gain map, when used to multiply the outputs of a model with no Gaussian gain (Multiply by propagated gain, *Figure 8a*) was sufficient to induce task performance benefits similar to Gaussian gain applied to the input (Propagated gain vs. Gaussian gain, *Figure 8c*). The median AUC across categories using the propagated gain map was 0.79, 95% CI [0.76, 0.84], compared to 0.71 [0.67, 0.72] in the distributed model. There was a small difference between the Gaussian gain and the effect of the propagated gain map -0.02 [–0.03, 0.01], within the 95% confidence interval for no difference. This difference could be attributed to changes in receptive field structure in the Gaussian gain condition, but we attribute it instead to differences between the propagated gain map and the effect of the Gaussian gain. The propagated gain manipulation was constructed from the average effective gain of units across all task stimuli. Because of this, the gain map did not exactly reproduce the effect of gain on an image-by-image basis.

To test the hypothesis that gain was necessary to account for the behavioral effect, we divided the final layer activations by the propagated gain map (Divide by propagated gain, *Figure 8a*). We found that the behavioral effect of an early gain was mostly reversed by this manipulation (Removed gain vs. Distributed, *Figure 8c*). The median AUC across categories after dividing out the propagated gain was 0.72, 95% CI [0.68, 0.75], compared to 0.71 [0.67, 0.72] in the distributed condition. Dividing by the propagated map did not perfectly reverse the effects of the full Gaussian gain. When compared to the distributed baseline, we found a median within category AUC advantage for Gaussian gain with division of 0.02 [0.01, 0.03]. These small residual differences are likely due to the combined effects of changes in spatial receptive field properties that are not reversed by the division of the propagated gain map.

Note that in the final readout of our model, we assumed that explicit spatial information was lost, as we averaged activations across the 7 × 7 convolutional units in the final pooling layer. However, some evidence in ventral temporal cortex suggests that there is spatial information available (*Schwarzlose*

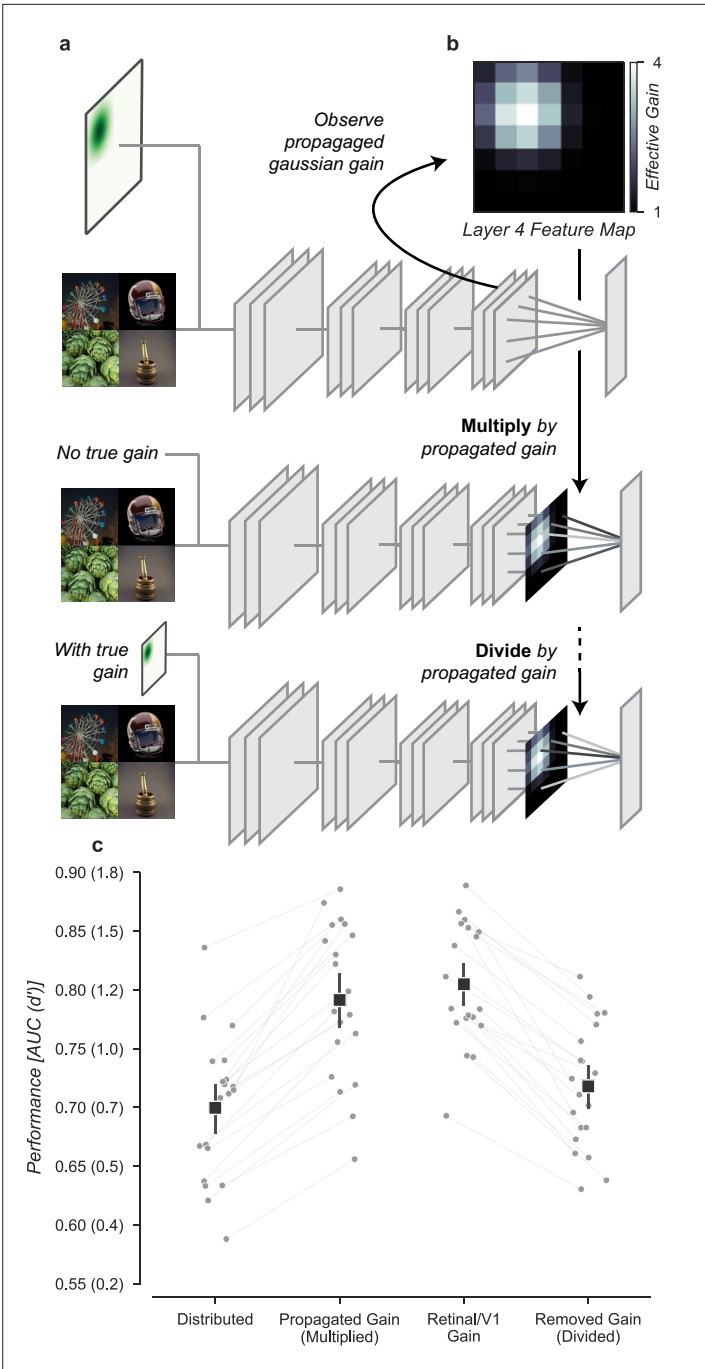

**Figure 8.** Gain is both necessary and sufficient to explain the improved task performance due to cued attention. (**a**) To test necessity and sufficiency of gain on performance, we propagated the effect of Gaussian gain through the model and measured the effective gain at the output layer. (**b**) We averaged the effective gain across features to obtain a 'propagated gain map'. To test sufficiency, we multiplied the output of a model with no true gain by the propagated gain map. To test necessity, we divided the output of a model with true gain by the propagated gain map. (**c**) Multiplying the output by the propagated gain recovered the effect of Gaussian gain, while dividing removed this effect, confirming that gain was both necessary and sufficient to account for the change in task performance. Grey markers show the individual category performance, black markers the median across categories (n=20) and error bars the 95% confidence intervals.

The online version of this article includes the following figure supplement(s) for figure 8:

*Figure 8 continued on next page*

*Figure 8 continued*

**Figure supplement 1.** Direct readout from the cued quadrant improves performance alone, with no additional improvement from gain.

**Figure supplement 2.** Gain propagation can account for changes in discrimination task performance due to Gaussian gain.

*et al., 2008*; *Carlson et al., 2011*), so we tested a model read out which retained spatial information and found that the necessary and sufficiency results did not show qualitative changes. A model trained to use the full $7 \times 7 \times 512$ output had marginally worse performance than the model built with the average encodings, achieving a median AUC across categories of 0.68 [0.63, 0.71] in the distributed condition and 0.78 [0.75, 0.82] in the focal condition with $4\times$ Gaussian gain at the first layer. We attribute the small difference in task performance compared to the average model to worse generalization: on the validation set the $7 \times 7$ model showed a median drop in AUC across categories of –0.02, range [–0.10, 0.00] compared to the average-pooled readouts.

We repeated the propagated gain manipulations in the $7 \times 7$ readout model to confirm the necessary and sufficiency results would not change when the model retained spatial information in the final readout. Both the necessity and sufficiency tests showed similar results when using the full output: the average increase in AUC when using the propagated gain map was 0.09, 95% CI [0.08, 0.10] for the full output model, compared to 0.09 [0.07, 0.10] for the average pooled model and the average change in AUC (compared to the distributed condition) when dividing out the propagated gain map from a model with Gaussian gain applied was 0.01 [-0.01, 0.02] for the full output model, compared to 0.02 [0.01, 0.03] for the average pooled model.

Improvements in task performance with a Gaussian gain could come from changes in signal discriminability, but also could come from the network being better able to suppress irrelevant visual information. That is, increasing the gain could act to strengthen signals from the relevant target and suppress signals from irrelevant locations. To see how much suppressing irrelevant visual information alone could improve task performance, we designed a neural network observer model which explicitly read out from the top-left $4 \times 4 \times 512$ quadrant of the layer 4 output, instead of the average pooled $1 \times 1 \times 512$ output. As expected, the task performance of this model with no additional gain is already elevated (Distributed, *Figure 8—figure supplement 1*), because the readout now implicitly acts as a form of spatial cueing. The performance of the 4×4 readout was still not at ceiling (ΔAUC between training validation set and distributed images = –0.07, 95% CI [-0.06,–0.09]). Thus, the performance enhancement due to the Gaussian gain appears to act similarly to an explicit manipulation which suppresses irrelevant information.

Theoretical considerations would suggest that moving receptive fields into the target quadrant should further improve performance even when the readout is already spatially specific, because these additional receptive fields can add new information (*Kay et al., 2015*; *Vo et al., 2017*). We found that this was not true for the amount of shift induced by the 4× Gaussian gain, chosen to match the magnitude of the human behavioral benefits of spatial attention. To demonstrate this, we applied a 4× Gaussian gain to the $4 \times 4$ readout model and found no further increase in performance beyond what was achieved by shifting the readout (Gaussian gain vs. 4 × 4 Readout, *Figure 8—figure supplement 1*). Gain applied to the output of the model also provided no additional benefit (Propagated Gain vs. 4 × 4 Readout, *Figure 8—figure supplement 1*), supporting the interpretation that the gain acts as a selection mechanism with no effect in the absence of irrelevant distractors.

Finally, the observer model solved a detection task where both criterion and sensitivity contribute to performance and we reported task performance as AUC to avoid confounding these factors. A more explicit test is to use a criterion-free discrimination task to evaluate the effects of gain on task performance. (*Figure 8—figure supplement 2a*). We therefore designed a category discrimination task in which the neural network observer model determined which of two composite grids included the target category at a specified location (always top-left). Baseline performance (Baseline, *Figure 8—figure supplement 2b*) was considered as the performance of the model when no information about which location is cued for discrimination was provided. Note that because the discrimination location always included the target and all other locations had equal probability of including the target category, chance performance was greater than 50%. To compute task performance, the previously trained fully connected category target readout was compared across the two

composite grids and the composite with the larger response chosen as the model's response. We then applied a Gaussian gain at the cued location and found that discrimination task performance improved in a similar manner to the detection task (Gaussian Gain, *Figure 8—figure supplement 2b*). Using the propagated gain manipulation, we confirmed that the gain was both necessary and sufficient for improvements in model task performance (Removed Gain vs. Baseline and Propagated Gain vs. Gaussian Gain, *Figure 8—figure supplement 2b*).

## Discussion

Human observers are more accurate when trying to detect or discriminate objects at a cued location. Our results demonstrate that this behavioral benefit can also be observed in a neural network model of visual cortex when a Gaussian gain is applied over the pixels of a 'cued' object. By modeling attentional modulation as gain at the earliest stage of the neural network, we were able to observe similar effects on spatial receptive fields to what is seen in human physiology. When using a gain strength set to match model and human performance enhancement during spatial attention, we documented shifts of receptive fields towards the center of the Gaussian gain field, shrinkage of receptive fields, and changes to the spatial structure in units at later stages of the model. These changes in model receptive field properties were similar in magnitude and characteristics to changes in single-unit (*Womelsdorf et al., 2006*; *Anton-Erxleben et al., 2009*) and population (*Klein et al., 2014*; *Vo et al., 2017*; *Fischer and Whitney, 2009*; *van Es et al., 2018*) receptive fields reported from physiological measurements.

To determine which, if any, of these changes to receptive field properties were the source of improved task performance in the model, we built a series of neural network observer models in which we isolated receptive field shifts, shrinkage, and structural changes from the direct effect of gain. To assess these changes in a way that could provide information about the human visual system, we matched the scale of the shifts, shrinkage, and structural changes to the effect size observed in the Gaussian gain model with the gain strength best matched to human performance. In the shift-only model, we re-wired units to move receptive fields without introducing gain and found that this produced no improvements in task performance. In the shrinkage model, we changed the size of units without changing their gain or position, and again found no improvements in task performance. In the receptive field structure model, we modified the sensitivity profile of individual receptive fields to mimic the effects of gain, without changing their gain, position, or size, but again found no improvements in task performance. It was only by applying a gain while keeping receptive field properties stable that we were able to reproduce the improvements in task performance.

Our results suggest that spatial gain implemented by neural populations in visual cortex can be sufficient to induce behavioral effects of attention for both detection and discrimination even without the concomitant changes in downstream receptive field properties. That is, increasing response magnitudes through gain changes can act to select relevant visual information when coupled with a max or soft-max pooling to suppress irrelevant visual information which has a lower response magnitude (*Lee et al., 1999*; *Pestilli et al., 2011*; *Hara et al., 2014*; *Pelli, 1985*). Although increasing gain can have downstream effects on receptive field properties such as changes in position, size and spatial structure, our results suggest that these may be secondary effects and only a consequence of applying gain, rather than the cause of the behavioral improvements as others have suggested (*Anton-Erxleben and Carrasco, 2013*; *Moran and Desimone, 1985*; *Anton-Erxleben et al., 2009*; *Sprague and Serences, 2013*; *Vo et al., 2017*; *Kay et al., 2015*; *Fischer and Whitney, 2009*; *Anton-Erxleben et al., 2007*). In addition, we found that gain had no additional impact when the readout was already spatially specific, reinforcing the interpretation that gain and selection of relevant information are intertwined.

We used an image-computable model of the computational steps from sensory input to decision making which allowed us to formally test hypotheses (*Gardner and Merriam, 2021*) about how different attentional mechanisms could impact task performance. In our case, the advantage of this approach is that the model architecture allowed us to examine how gain at the earliest stages of processing causes changes in spatial receptive field properties: any time a gain occurs in an asymmetrical manner across a receptive field, downstream units will experience an apparent shift as well as shrinkage or expansion. We know from the large literature exploring the physiology of attention that receptive field shifts are correlated with spatial attention (*Anton-Erxleben and Carrasco, 2013*;

*Anton-Erxleben et al., 2009*; *Anton-Erxleben et al., 2007*; *Vo et al., 2017*; *Kay et al., 2015*; *Fischer and Whitney, 2009*; *Womelsdorf et al., 2006*). Several authors have proposed that enhanced task performance is a result of increasing information capacity by reducing spatial uncertainty about position (*Kay et al., 2015*) or enhancing discriminability (*Vo et al., 2017*). However, if changes in spatial receptive field properties are the consequence of gain changes (*Klein et al., 2014*; *Compte and Wang, 2006*), then it raises the question of whether these receptive field changes actually help to improve task performance. Our modeling approach allowed us to examine the theoretical impact of each change associated with gain systematically and quantify that these provided no benefit to detection or discrimination task performance. Nevertheless, previous modeling work has demonstrated that shift and shrinkage, in particular, can increase the resolution and redundancy of receptive field coverage (*Kay et al., 2015*; *Vo et al., 2017*; *Theiss et al., 2022*). Our results differ along two important dimensions: first, we scaled the magnitude of shift, shrinkage, and sensitivity changes to those induced by Gaussian gain (at a gain strength that was matched to human performance) and these were in general equal to or smaller in magnitude to what was observed in these previous papers. This leaves open the possibility that larger shifts or shrinkage could have a more direct impact on task performance. We also note that our results were based only on simulations for category detection and discrimination tasks. It may be the case that other tasks which depend more directly on spatial coding, such as judgements of visual position, could exhibit larger benefits from shifts and shrinkage of receptive fields (*Kay et al., 2015*; *Vo et al., 2017*).

Whether our conclusions can generalize to the behavior of attentional gain in biological neural circuits is limited both by how well the neural network observer model approximates the functioning of those neural circuits and by the model's ability to predict behavior. There are several reasons to suggest that the model captures relevant properties of both object recognition and the primate visual system. We chose to analyze a CNN whose architecture was designed to reflect the primate visual system. This has been evaluated by comparing the similarity of CNN unit activity against measurements of single unit activity in the primate visual cortex (*Schrimpf et al., 2018*). After training, the image features that the CNN units become selective for align closely with those that activate single units in visual cortex (*Yamins et al., 2014*; *Carter et al., 2019*). In addition, the designers of the architecture we used (CORnet), *Kubilius et al., 2018* optimized for 'core object recognition', detecting a dominant object during a viewing duration of natural fixation (100–200ms) in the central visual field (10 deg). We re-used core object recognition in our human object detection task and projected our composites in a 10 degree square aperture to obtain similar perceptual characteristics. In the analysis of our task we showed that distributed performance was similar for humans and the CNN at a stimulus presentation of 65ms, confirming that the intended design of CORnet generalized to the new dataset and task that we used.

While CORnet was designed to map individual visual cortex regions onto the different layers of the CNN, it differs from the visual system in that it is a completely feed-forward model. It is well-known that the visual system has recurrence both within and between visual areas (*Felleman and Van Essen, 1991*). Computational modeling has suggested that recurrence can affect how gain and additive offsets change down stream receptive field location and size, in particular enhancing these effects beyond receptive field boundaries (*Compte and Wang, 2006*). These considerations suggest that more realistic models could have even stronger downstream effects on spatial receptive field properties then what we have documented in a purely feed-forward network. In computational models, recurrent connections are often unfolded into feed-forward layers, effectively making a recurrent model a deeper convolutional model (*Nayebi et al., 2018*). Although we did not test deeper architectures in our analysis, we expect that the general principles we described should hold for models with more layers and therefore also for models with recurrent connections. An intriguing follow-up direction would be to extend the modeling described here to reaction time tasks, where a recurrent architecture allows for modeling of temporal dynamics and where diffusion models have been found to provide a useful parameterization of how bottom-up and top-down signals contribute to sensory responses over time (*Kay and Yeatman, 2017*).

CORnet is also missing many intermediate areas of the visual system (notably area V3) (*Wandell and Winawer, 2011*) as well as an explicit gain control mechanism such as divisive normalization (*Carandini and Heeger, 2012*) which might account for the large gain necessary in our model to produce human-like performance enhancements. These differences mean that the exact strength of

the gain signal we observed cannot be mapped directly onto physiology. In particular, while we apply gain at the earliest stage of the model, we do not wish to imply that such a large gain is seen with attention in the LGN inputs to V1 (*O'Connor et al., 2002*). Nor do we imply that the gain in various stages of our model should directly map on to the gain observed in physiological measurements, which have tended to highlight larger gain changes in intermediate areas like V4 and MT (*Treue and Martínez Trujillo, 1999*; *McAdams and Maunsell, 1999*; *Moore and Armstrong, 2003*) compared to earlier areas. Instead, in our model, the 4× gain should be interpreted as both an explicit increase in gain as well as an implicit gain due to the effects of normalization (*Reynolds and Heeger, 2009*; *Carandini and Heeger, 2012*). While normalization models have traditionally been studied in single layer models, our work extends this general approach to consider downstream effects of gain on RF properties. We assessed how these effects might interact in our CNN by demonstrating that a physiologically plausible gain of 1.1×, when accentuated by a divisive gain control mechanism (*Kaiser et al., 2016*; *Carandini and Heeger, 2012*) and amplified across multiple visual areas (many of which are not included in the CORnet model), could have produced the magnitude of effects necessary for human-level improvements in task performance. This smaller gain is more consistent with neural recordings in primates, where gain changes on the order 20–40% (1.2–1.4×) have been measured (*Motter, 1993*; *Luck et al., 1997*; *Treue and Martínez Trujillo, 1999*).

We chose to model gain at the earliest possible point in the system to understand how signal changes propagate through the visual hierarchy and modify receptive field structure. Physiological measurements have found evidence for early gain (*McAdams and Maunsell, 1999*; *Motter, 1993*; *Luck et al., 1997*), but it is equally possible that the gain is applied at a late stage close to decision making and signal gains early in visual cortex are a result of backward projections to these areas (*Buffalo et al., 2010*; *Moore and Armstrong, 2003*). The propagated gain analysis confirms that gain signals with spatial specificity arriving at later stages in processing (*Moore and Armstrong, 2003*) would have similar effects on task performance.

To solve the demands of goal-directed visual attention, the human brain has multiple potential mechanisms available. To select for relevant and suppress irrelevant information, sensory responses can be amplified or the tuning of neurons and populations can be shifted to enhance some signals at the cost of others. In addition, these bottom-up sensory changes can be combined with changes in how sensory representations are read out or communicated to downstream regions. In biological systems, these mechanisms are intertwined: as we have shown, changes to early sensory signals will have complex effects on the later stages that are used for readout. In an idealized model, the changes that would have the most effect on the readout would be computed by approximating their gradients on the decision axis (*Lindsay and Miller, 2018*). However, these gradients are typically computed in models through back-propagation (*Rumelhart et al., 1986*), and it is not known whether or how similar gradients can be computed in biological systems. Here, we have shown using a state-of-the-art model of the visual system that when the neural network observer is matched with human performance during spatial attention some mechanisms can improve task performance, while others cannot. In the limit, shift, shrinkage, and tuning changes in receptive fields must have an impact on sensory representations and therefore on performance. But our results show that in a neural network model and at the scale expected in the primate visual system during goal-directed behavior, these are not sufficient to produce the expected effects of spatial attention on task performance. Instead, gain combined with a nonlinear selection mechanism meets the demands imposed by goal-directed visual attention. New techniques that allow for targeting interventions to defined populations of neurons raise the possibility of manipulating gain and top-down signaling to determine the effect on downstream neural response properties and behavior. Such interventions would allow for testing the main prediction of our model: that spatial visual attention relies primarily on changes in gain and not concomitant downstream effects to spatial receptive field properties.

## Materials and methods
### Human observers
Seven observers were observers for the experiments (1 female, 6 male, mean age 22 y, range 19–24). All observers except one (who was an author) were naïve to the intent. No observers were excluded during the initial training sessions (see eye-tracking below). Observers completed 1600 trials in two

60-min sessions. Observers wore lenses to correct vision to normal if needed. Procedures were approved in advance by the Stanford Institutional Review Board on human participants research and all observers gave prior written informed consent before participating (Protocol IRB-32120).

## Hardware setup for human observers

Visual stimuli were generated using MATLAB (The Mathworks, Inc) and MGL (*Gardner et al., 2018*). Stimuli were displayed at 60 cm viewing distance on a 22.5 inch VIEWPixx LCD display (resolution of 1900x1200, refresh-rate of 120 Hz) and responses collected via keyboard. Experiments were performed in a darkened room where extraneous sources of light were minimized.

Eye-tracking was performed using an infrared video-based eye-tracker at 500 Hz (Eyelink 1000; SR Research). Calibration was performed at the start of each session to get a validation accuracy of less than 1 degree average offset from expected, using a thirteen-point calibration procedure. During training, trials were initiated by fixating the central cross for 0.5 s and canceled on-line when an observer's eye position moved more than 1.5 degree away from the center of the fixation cross for more than 0.3 s. Observers were excluded prior to data collection if we were unable to calibrate the eye tracker to an error of less than 1 degree of visual angle or if their canceled trial rate did not drop to near zero. All observers passed these criteria. During data collection the online cancellation was disabled and trials were excluded if observers made a saccade outside of fixation (>1.5 deg) during the stimulus period.

## Experimental design

We compared the ability of humans and neural networks to detect objects in a grid of four images covering 10 degrees of visual angle (224 px). Given a grid of images, the observers were asked to identify whether or not a particular target category was present. On half of the trials we gave observers prior information telling them which of the four grid locations could contain the object (100% valid cue). This focal condition was compared with a distributed condition, in which no information was provided about which grid location could contain the target object. For humans, the prior in the focal condition was a spatial cue, a visual pointer to one corner of the grid. For the neural network, the prior for the focal condition was implemented by a mechanistic change in the model architecture, which differed according to the model of attention being tested. For the neural network, note that in the distributed condition the model is analogous to one in which the focal cue is implemented by a Gaussian of infinite width.

To verify that our results were not specific to detection, we also examined the ability of a neural network observer model to perform a category discrimination task. To perform the discrimination, we compared the classifier outputs from two composite grids. These grids were constructed such that one of the two grids always contained an image of the target category (A) in the top-left location and the other contained an image from the non-target category (B). The remaining distractors images were randomly sampled from the A and B categories with 50% probability. In the focal cue condition, the model architecture was modified to implement a model of attention.

## Stimuli: object detection task

In the object detection task, the stimuli presented to both humans and the neural network observer model were composed of four base images arranged in a grid (henceforth a 'composite grid'). Each base image contained an exemplar of one of 21 ImageNet (*Deng et al., 2009*) categories. Composite grids always contained images from four different categories. The base images were cropped to be square, and resized to $122 \times 122$ pixels, making each composite grid $224 \times 224$ pixels. We pulled 929 images from each of 21 ImageNet categories: analog clock (renamed to 'clock'), artichoke, bakery (renamed to 'baked goods'), banana, bathtub, bonsai tree (renamed to 'tree'), cabbage butterfly, coffee, computer, Ferris wheel, football helmet, garden spider (renamed to 'spider'), greenhouse, home theater, long-horned beetle (renamed to 'beetle'), mortar, padlock, paintbrush, seashore, stone wall, and toaster. These base images were usually representative of their category. However, many included other distracting elements (people, text, strong reflections, etc). Two authors (KF and DB) selected 100 base images for each category absent of distracting elements (low-distraction base images) to be used for the human task. From these low-distraction base images, we set aside 5 to use as exemplars when introducing the category to human participants.

**Table 1.** Category pairs for the discrimination task.

| Pair | Category A | Category B |
|------|-----------|-----------|
| 0 | Ferris wheel | Analog clock |
| 1 | Artichoke | Bakery |
| 2 | Banana | Bathtub |
| 3 | Cabbage butterfly | Coffee |
| 4 | Computer | Football helmet |
| 5 | Garden spider | Greenhouse |
| 6 | Home theatre | Long-horned beetle |
| 7 | Mortar | Padlock |
| 8 | Paintbrush | Seashore |
| 9 | Stone wall | Toaster |

To create the human stimulus set we generated composite grids for each of the 20 target categories. Each category required 80 composite grids: 40 including target objects and 40 without. We therefore needed 40 base images from the target category and 280 $(3 \times 40 + 4 \times 40)$ base images from the non-target categories. We sampled all images from the low-distraction base images. Targets were placed 10 times in each of the four corners.

The neural network observer model was trained and tested on an expanded stimulus set. We set aside 50 base images for each category to train the linear classifiers (see Linear Classifiers, below). The approach was otherwise identical to that described above, but 829 composite grids were created with a target and 829 without, and the composites were assembled from the full set of 929 base images. Because CNN models are translation invariant we formed all target composites with the target image in the NW corner, to simplify analysis.

## Stimuli: category discrimination task

The stimuli in the category discrimination task were also composite grids of four images. However, these composites were constructed to only include images from a target pair of categories (called 'A' and 'B' and generated from 20 of the 21 ImageNet categories, as displayed in *Table 1*). Pairs of composites were generated, consisting of an 'A' stimulus and a 'B' stimulus with the corresponding category in the top left target grid position. The other three locations were filled with distractor images sampled pseudorandomly from the A or B category. Target images were not repeated across composites, but did appear in other stimuli as distractors. We generated 900 images per category pair, 450 with an A target and 450 with a B target.

## Human object detection task

Human observers performed blocks of trials in which they had to report the presence or absence of a specified category in composite grids. At the start of each block we showed the human observers the words 'Search for:' followed by the name of the current target category (*Figure 1a*, Category). They were then shown five held-out (i.e. not shown in the task) exemplar base images to gain familiarity with the target category (*Figure 1a*, Examples) and advanced through these with a self-paced button click. This was followed by individual trials of the task. At all times, a fixation cross (0.5 deg diameter, white) was visible at the center of the screen in front of a black circle (1 deg diameter). This fixation region obscured the center of the composite grid, but made maintaining fixation easier for observers. At the start of each trial, the pixels of the current composite grid were scrambled to create a luminance-matched visual mask. This was displayed until an observer maintained fixation for 0.3 s (*Figure 1a*, 'Fixation'). Once fixation was acquired a cue was shown for 0.75 s, informing the observer about whether the trial was focal (in which case the possible target location was indicated) or distributed (four possible target locations indicated). The focal cue was a 0.25 deg length white line pointing toward the cued corner of the grid. The distributed cue was four 0.25 deg length white lines pointing toward all four corners of the grid. Distributed and focal cues were presented in pseudorandomized order throughout each block. The cue was followed by a 0.75 s inter-stimulus interval (*Figure 1a*, Delay) before the composite grid (10 × 10 deg) was shown for either 1 (8.3 ms), 2 (16.7), 4 (33.3), 8 (66.7), 16 (133.3), or 32 (266.7) video frames (*Figure 1a*, Stimulus). The mask then replaced the stimulus and observers were given 2 s to make a response (*Figure 1a*, Response), pressing the '1' key for target present or the '2' key for absent. Feedback was given by changing the fixation cross color to green for correct and red for incorrect until the 2 s period elapsed. A 0.25 s inter-trial interval separated trials.

**Table 2.** CORnet-Z structure.

Average receptive field (RF) full-width at half-maximum (FWHM) is measured using ellipses fit to the backpropagated gradients of units in a convolutional layer with respect to the input image pixels. 22.4 pixels corresponds to one degree of visual angle (*Kubilius et al., 2018*).

| | Layer Type | Kernel Size | Output Shape | FWHM (px, deg) |
|---|---|---|---|---|
| Input | | | 224 × 224 × 3 | |
| V1 Block | conv, stride = 2 | 7×7 | 112 × 112 × 64 | 11 (0.5) |
| | ReLU | | 56 × 56 × 64 | |
| | max pool | 2×2 | 56 × 56 × 64 | |
| V2 Block | conv | 3×3 | 56 × 56 ×128 | 26.8 (1.21) |
| | ReLU | | 28 × 28 ×128 | |
| | max pool | 2×2 | 28 × 28 ×128 | |
| V4 Block | conv | 3×3 | 28 × 28 × 256 | 55.6 (2.52) |
| | ReLU | | 14 × 14 × 256 | |
| | max pool | 2×2 | 14 × 14 × 256 | |
| IT Block | conv | 3×3 | 14 × 14 × 512 | 111.4 (5.06) |
| | ReLU | | 7 × 7 × 512 | |
| | max pool | 2×2 | 7 × 7 × 512 | |
| Encodings | avg. pool | | 1 × 1 × 512 | |

Observers completed one training block (the 'tree' category) as practice before data collection began. They then completed each category block (40 focal trials with 20 target present and 20 target absent, and 40 distributed trials with 20 target present and 20 target absent) before moving on to the next category. Block order was pseudo-randomized for each observer. Each block took about five minutes to complete and a break was provided between blocks, as needed. In total, the experiment took about 2 hr, split into two 1-hr sessions on different days.

## Neural network observer model

We modeled the ventral visual pathway using CORnet-Z, a convolutional neural network (CNN) proposed by *Kubilius et al., 2018*. The model consists of four convolutional layers producing feature maps of decreasing spatial resolution (*Table 2*). The model which we used was pre-trained on ImageNet by the original authors (*Kubilius et al., 2018*). At the last convolutional layer we took the average over the spatial dimensions of each feature map to create the neural network's representation (512-dimensional vector) of the input image.

## Linear classifiers: object detection task

To allow the neural network observer model to perform an object detection task, we trained a set of linear classifiers on the model output to predict the presence or absence of each of the twenty target categories. Each of these fully connected layers received as input the (512-dimensional) feature output from the CNN and projected these to a scalar output. Weights were fit using logistic regression with L2 regularization, using *scikit-learn* and the *LIBLINEAR* package (*Pedregosa et al., 2011*). We trained the classifiers on a held out set of base images not used to generate the task grids, using 50 images with the target present and 50 images with the target absent. Training was evaluated on an independent validation set of 100 images, median AUC 0.90, range [0.77, 0.96].

To test model performance in the detection task the observer model was presented with each of the composite grids in the full image set and the output of the target category's classifier was computed. We report the model's area under the curve (AUC) as a measure of performance. The AUC is computed from the area under the curve defined by plotting the false positive rate against the true positive rate across the full range of possible thresholds (0–1). We used the *scikit-learn* implementation

to calculate the AUC for each model. The AUC can be interpreted as the average probability that target images will be ranked higher by the logistic regression compared to non-target images, with a value of 0.5 indicating chance performance and a value of 1 indicating perfect discrimination. In a signal detection framework, an AUC of 0.75 corresponds approximately to a d' of 1.

## Linear classifiers: category discrimination task

To allow the neural network observer model to perform a category discrimination task, we repeated the linear classifier training described above, adding a final step in which the classifier outputs were compared for two composites. The composite grid producing a higher output was marked as containing the target category. The classifiers were trained on a held out set of base images not used to generate the task grids. Because this task is criterion-free, we report the model's accuracy as a measure of performance. Note that even in the distributed condition the model performance exceeds chance: this is because in any set of category pair composites the proportion of grid positions with a target will always be higher when the target image is fixed to one category. On average across images the proportion of A images in the A targets will be 2.5/4 $(1 + 0.5 + 0.5 + 0.5)$.

## Spatial attention: Gaussian gain model

To introduce Gaussian gain as a mechanism for spatial attention, we multiplied the pixel intensity of the input image at row $r$ and column $c$ by the magnitude of a 2-dimensional Gaussian, using the following equation:

$$g_{r_0,c_0,\sigma,\beta}(r,c) = (\beta - 1) \exp\left(-\frac{(r - r_0)^2 + (c - c_0)^2}{2\sigma^2}\right) + 1 \tag{2}$$

where $r_0$ and $c_0$ set the row and column location for the center of the gain field and $\beta$ controls the strength, that is the multiplicative factor at the peak of the Gaussian. The Gaussian was centered in the cued quadrant and $\sigma$ was set to 56 pixels (approx 2.5 degrees). We explored four values of $\beta$: 1.1, 2, 4, and 11.

## Quantifying the effects of gain on receptive fields and activations

To reduce computational requirements we randomly sampled 300 units per layer (1200 total units) for receptive field analysis, with higher density near the attended locus.

To determine the location and size of the receptive field of each CNN unit, we computed the derivative of their activation with respect to the pixels in the input image. This derivative was taken across a batch of 40 task images evenly distributed across categories. The magnitude of derivatives with respect to the red, green and blue channels were summed to create a sensitivity map. Receptive field location and size were estimated by fitting a 2D Gaussian distribution. The fit was performed by treating the sensitivity map as an unnormalized probability distribution and choosing the Gaussian with the same mean and covariance matrix as that distribution. Receptive field location was measured as the mean of the Gaussian fit. We report the full-width at half-maximum for the receptive field size.

To measure the effect of gain on the activation and information content of CNN units, we computed the effective gain and the change in AUC across the sampled units. We defined effective gain as the ratio between the standard deviation of a unit's activity after applying an attention mechanism compared to before. We computed the effective gain across all features and all stimuli. To compute the change in AUC, we measured the average change along the prediction layers' decision axes for each feature map location in layer 4 between the distributed and focal conditions. More specifically, for each category and each location in the $7 \times 7$ feature map, we passed the 512-dimensional encoding vector onto that category's prediction layer just as we did for the 512-dimensional vector after average pooling. This resulted in two distributions of confidence scores along the prediction layer's decision axis (one each for target present and absent), the AUC of which describes the relative amount of information contained in that feature map location pertaining to discrimination of target present and absent conditions. We then took the difference of AUCs between focal and distributed conditions averaged across categories in each location.

## Nonlinear normalization

In order to test the ability of 'winner-take-all' normalization to amplify small gains, we isolated the first layer of the CNN, and applied nonlinear normalization with exponent $\xi$. More precisely, if the output

feature map of the first layer had size $M$ rows by $N$ columns by $C$ channels and activations $a_{ijc}$, we calculated the normalized outputs:

$$b_{ijc} = \frac{\sum_{k,l,d=1}^{M,N,C} |a_{kld}|}{\sum_{k,l,d=1}^{M,N,C} |a_{kld}|^{\xi}} a_{ijc}^{\xi}. \tag{3}$$

To measure the resulting amplified gain, we applied a small Gaussian gain between $1\times$ and $1.1\times$ to the input image in the same manner as in the full Gaussian gain model. We then measured the ratio of average effective gain for units contained entirely within the gain field against the average effective gain of units entirely outside the attention gain field, for various values of $\xi$.

## Spatial attention: shift-only model

In the Gaussian gain model, we applied the gain at layer 1 and observed changes in the model's detection performance at the output layers. We took a parallel approach here to design a model that could mimic the receptive field shifts at layer 4 (induced by gain at layer 1) while producing no systematic effect on response gain. To cause the layer 4 units to observe different parts of the input image, we shifted the connections between pixels in the input image and first layer. We preserved all other connections, so layer 4 units of the neural network continued to receive information from the same layer 1 units.

To obtain the size of the necessary connection shifts we created a 'shift map' in input image space by measuring the distance and direction that layer 4 units moved when the Gaussian gain was applied. To make this measurement, we took each input image pixel location $(r, c)$ and calculated the average receptive field shift of the 20 sampled layer 4 units with the closest receptive field centers without attention. Because we used a sampling procedure and not the full set of layer 4 units we weighted the sampled units by their Euclidean distance from the target pixel. To reduce noise in the shift map, we applied a Gaussian blur with $\sigma = 8$ pixels. Using the shift map, we then re-assigned the connections from the input image to the layer 1 units. The simplest way to to implement this involved swapping the activation of each layer 1 unit with the activation of the unit at its shifted location. For example, if unit $(75, 75)$ was shifted by $(-10, -10)$ we assigned it the activation of the unit at $(65, 65)$. To deal with decimal shifts we performed linear interpolation using neighboring units.

## Spatial attention: receptive field structure

In the receptive field structure model we aimed to mimic the spatial tuning changes induced by the Gaussian gain at a particular layer but without changing the effective gain of units. To accomplish this, we first computed the true gain propagated to the target layer $L$ by scaling the Gaussian gain map to the size of layer $L-1$'s feature map. With this change alone the weights of units closer to the locus of attention are scaled more than the weights farther from the locus, introducing differential gain. To avoid a change in the overall scale of units' weights, we re-scaled the kernel to match the L2-norm (sum-of-squares) of the original kernel weights.

To summarize, suppose that layer $L-1$'s feature map is $t$ times the size of the input image so that a unit at row $r$ and column $c$ of the layer $L-1$ feature map has an effective effective gain of $g_{tr_0, tc_0, t\sigma, \beta}(tr, tc)$ under the Gaussian gain model. Then if $w \in \mathbb{R}^N$ is the original weight vector of a unit in the unraveled convolution at layer $L$ whose input vector $a \in \mathbb{R}^N$ contains the activations of post-ReLU units of layer $L-1$, and if the row-column positions in the $L-1$ feature map of the unit described by $a_i$ is $(r_i, c_i)$, then the replacement weight vector in the sensitivity shift model is given by the vector $w' \in \mathbb{R}^N$, whose entries are:

$$w_i' = \left( \frac{\sum_{i=1}^{N} w_i^2}{\sum_{i=1}^{N} w_i^2 g_{tr_0, tc_0, t\sigma, \beta}(tr_i, tc_i)^2} \right)^{1/2} w_i, \tag{4}$$

## Spatial attention: shrinkage model

In the shrinkage model we aimed to mimic the receptive field size changes observed at layer 4 under Gaussian gain, without causing changes in receptive field location or gain. To achieve this, we assigned a shrinkage factor to each layer 4 unit and rewired its connections to layer 3 accordingly.

Shrinkage factors $f_\beta(d)$ were determined by the distance $d$ between the locus of attention in input image space and the unit's spatial location in the feature map projected back onto the input image. This distance was converted to a shrinkage factor by a function chosen to model the properties of the receptive field size change pattern observed under $11x\times$ Gaussian gain at layer 4 (**Figure 3e**), namely

$$f_\beta(d) = 1 - \beta \exp\left(-2.44\frac{d^2}{112^2}\right) \cos\left(2.89\frac{d^2}{112^2}\right) \tag{5}$$

where $\beta$ determines the overall strength of the effect, and ranged from 0.1 to 0.4 in our analyses. A shrinkage factor of 0 indicated no change in receptive field size, while a shrinkage factor of 1 indicated shrinkage to zero radius.

Given a shrinkage factor, we re-weighted the connections of each layer 4 unit to produce an approximate shrunken convolutional kernel for that unit. The linearity of convolution provides an equivalence between re-weighting connections from layer 3 to layer 4 and replacing those connections with new ones to units in a virtual continuous layer 3 feature map formed by linear interpolation between activations in the true layer 3 feature map. We therefore were able to calculate the new weights for each layer 4 unit based on a length-9 array of floating-point locations on the layer 3 feature map (all CORnet-Z kernels are $3 \times 3$). Given the original wiring locations $x_i, y_i, i = 1, ..., 9$ for a unit with distance $d$, the new location corresponding to input $i$ was chosen to be

$$x_i' = f(d)x_i - (1 - f(d))\left(\frac{1}{9}\sum_{j=1}^{9}x_j\right) \tag{6}$$

and similarly for $y_i'$. Using the linearity of convolution, each new virtual input location $(x_i', y_i')$ is equivalent (for a linearly interpolated feature map) to a weighted combination of connections to the four feature map locations surrounding $(x_i', y_i')$, calculated by rounding $x$ and $y$ coordinates up or down. The resultant 36 ($9 \times 4$) connections were then simplified by combining connections from the same layer 3 unit to yield a re-weighted convolution kernel.

### Spatial attention: Gain-only model

We designed a model which could effect gain without receptive field shift by flattening the gain in the cued quadrant. Receptive field shift occurs when there is a differential gain across the receptive field of a unit. To get rid of this, you can put a flat gain over the cued quadrant. This naive approach has the problem that units that overlap two quadrants will still shift and shrink according to the strength of the gain. To prevent these units from shifting in a manner correlated to the gain we separated the CNN feature maps into four parts corresponding to the four image quadrants, ran the model forward with zero padding around each quadrant, and then concatenated the results back together. This ensured that each unit experienced a flat gain across its inputs and that as gain increased units near the quadrant boundaries did not experience gain-dependent receptive field shift or shrinkage.

### Necessary and sufficient test

To obtain a propagated gain map in the final layer output we applied the Gaussian gain to the start of the neural network observer model and measured the average effective gain of the $7 \times 7$ layer 4 output units across a representative sample of images. We call this the 'propagated gain map', since it represents the effect of the input gain on the output layers. We tested necessity by dividing the network output by the map for a model with gain applied and we tested sufficiency by multiplying the outputs from a no-gain model.

### Readout from target quadrant

To test the behavior of the neural network observer model with spatially-specific readout from the last convolution layer (Layer 4), we masked the output of that layer to the linear prediction layers in the object detection task. To apply the mask, we zeroed activations of units outside the top-left $4 \times 4 \times 512$ of layer 4 (full dimensions $7 \times 7 \times 512$). The same linear prediction layers and stimuli were used as in the necessary and sufficient test, and the same four conditions were tested: no gain, early Gaussian gain, and with a propagated gain map applied and divided out at layer 4.

## Behavioral analysis

We analyzed the human behavioral data by binning trials according to their duration and computing sensitivity $d'$ from the equation:

$$d' = Z(H) - Z(FA) \tag{7}$$

where $Z$ is the inverse of the cumulative normal distribution and $H$ and $FA$ are the hit and false alarm rate, respectively. We fit a logarithmic function to the $d'$ data using the equation:

$$d'(t) = \alpha * \log(\kappa t + 1) \tag{8}$$

where $t$ is the stimulus duration and $\alpha$ and $\kappa$ are parameters that control the shape of the logarithmic function.

To compare human and model performance, we can also convert between $d'$ and the area under the curve (AUC) by the equation:

$$d' = \sqrt{2}Z(AUC) \tag{9}$$

## Confidence intervals

All error bars are calculated by bootstrapping the given statistic with $n = 1000$ and reported as the 95% confidence interval.

## Acknowledgements

We thank Josh Wilson for help with data collection and Eline Kupers and Maggie Henderson for early discussions.

## Additional information

### Funding

| Funder | Grant reference number | Author |
| --- | --- | --- |
| Washington Research Foundation | Postdoctoral Fellowship | Daniel Birman |
| Research to Prevent Blindness | | Justin L Gardner |
| Lions Clubs International Foundation | | Justin L Gardner |
| Hellman Fellows Fund | | Justin L Gardner |
| National Eye Institute | T32EY07031 | Daniel Birman |

The funders had no role in study design, data collection and interpretation, or the decision to submit the work for publication.

### Author contributions

Kai J Fox, Conceptualization, Software, Formal analysis, Investigation, Visualization, Methodology, Writing – review and editing; Daniel Birman, Conceptualization, Formal analysis, Supervision, Investigation, Writing – original draft, Writing – review and editing; Justin L Gardner, Conceptualization, Supervision, Funding acquisition, Project administration, Writing – review and editing

### Author ORCIDs

Daniel Birman ⓘ http://orcid.org/0000-0003-3748-6289
Justin L Gardner ⓘ http://orcid.org/0000-0003-2220-5050

## Ethics

Human subjects: Procedures were approved in advance by the Stanford Institutional Review Board on human participants research and all observers gave prior written informed consent before participating (Protocol IRB-32120).

## Decision letter and Author response

Decision letter https://doi.org/10.7554/eLife.78392.sa1
Author response https://doi.org/10.7554/eLife.78392.sa2

---

# Additional files

## Supplementary files
• MDAR checklist

## Data availability

The images and composite grids used in this study as well as the code necessary to replicate our analyses are available in the Open Science Framework with the identifier https://doi.org/10.17605/OSF.IO/AGHQK.

The following previously published dataset was used:

| Author(s) | Year | Dataset title | Dataset URL | Database and Identifier |
|---|---|---|---|---|
| Fox KJ, Birman D, Gardner JL | 2020 | Gain, not changes in spatial receptive field properties, improves task performance in a neural network attention model | https://doi.org/10.17605/OSF.IO/AGHQK | Open Science Framework, 10.17605/OSF.IO/AGHQK |

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
