## [Editor Report]

This manuscript combines human behavioral experiments using a categorization task and a convolutional neural network model to test different mechanisms that may support attention-related improvements in perception. Through carefully controlled manipulations of computational architecture and parameters, the authors dissociate the effects of tuning gain vs. tuning shifts. They conclude that increases in gain are the primary means by which attention improves behavioral performance.

---

## [Decision Letter]

**Decision letter after peer review:**

Thank you for submitting your article "Behavioral benefits of spatial attention explained by multiplicative gain, not receptive field shifts, in a neural network model" for consideration by *eLife*. Your article has been reviewed by 2 peer reviewers, and the evaluation has been overseen by a Reviewing Editor and Floris de Lange as the Senior Editor. The following individuals involved in the review of your submission have agreed to reveal their identity: Albert Compte (Reviewer #1); Kendrick N Kay (Reviewer #2).

Essential revisions:

We were all very enthusiastic about the approach and the findings. During our discussions, we collectively focused on several issues regarding the generality of the conclusions and the methods used to arrive at them. Below is a list of the main issues:

1) The Detection vs discrimination. Having at least one other task would allow you to determine if the gain story is specific to detection or if the type of modulation depends on the task. In either case, results from another task would be interesting and would greatly improve the generality and theoretical impact of the claims.

2) Justifying the spatial invariance of the readout mechanism and potentially testing other schemes.

3) What happens when gain isn't just injected into "V1", but into other layers? For example, Lindsay has shown that this matters in the context of CNNs performing attention tasks, and the rationale for injecting a change so early in the system isn't entirely clear.

4) Others, like Compte and Wang 2006, have also explored this question using other architectures. Given that the CORnet-z model does not include recurrent dynamics within each layer, it will be important to discuss and consider the implications of the chosen network and implications for the generality of the conclusions.

*Reviewer #1 (Recommendations for the authors):*

1) The model requires a more detailed presentation in the text (now lines 93,94). A succinct description of how it is built should be provided so the reader does not have to look up the references for a simple, broad-picture understanding. It is important to mention that the model has been built and trained previously based on the same set of images that will be used here, and the weights are available, so the only further training done here is the linear classifier on the model outputs to detect image categories upon presentation of "composite" grids. It took a while for me to understand this and to see the difference between base images and composite images. There is also a confusing thing about the neural model in Figure 2d: it pictures the ReLU occurring after the max pooling step, but in Figure 1 of Kubilius et al. (2018) it occurs before the max pooling step. The sentence ("Unit activations were measured after the convolution, prior to the max pooling step") in the caption of Figure 2 is also confusing: how is unit activity being measured before the ReLU non-linearity that transforms inputs into output rates?

2) The manuscript emphasizes 3 different RF modulations: scaling, shift, and shrinkage, but then only two of them are specifically isolated with network simulations. RF shrinkage is not addressed. This is not really a problem, but it would be nice to have full "symmetry" in the manuscript. Is there a specific reason why this modulation can not be assessed to make it more parallel to the rest?

3) References are required in lines 145 ("…behavioral benefits of attention") and 250 ("…as others have suggested").

4) In general, scatter plots in the figures should indicate what are the individual data points presented. I think in some cases are units, in other cases they are categories. This should be clearly indicated.

*Reviewer #2 (Recommendations for the authors):*1. On the issue of the readout.As mentioned in the public review, it appears that the conclusions are heavily dependent on the choice of readout mechanism. Thus, it seems a bit premature at this point to make strong general conclusions about which tuning properties are critical for behavioral improvements. Many recent studies have shown spatial tuning (i.e. large but limited RF sizes) in the high-level visual cortex; hence, in the neural network observer model, it is not clear why the readout (linear classification) is performed on unit activity that has been summed fully across visual space.

Now, it is an open interesting empirical question whether perceptual decisions from the visual system are based on fully spatially invariant units. But note that if that were the case, isn't one counterintuitive implication that our perceptual decisions can be corrupted by visual stimuli far away from the relevant decision zone?

If we entertain the possibility that the readout can be from units that have spatial specificity, then it remains to be seen what sort of modeling outcomes there might be.

Even though the authors show, in the context of their particular model, that tuning shifts do not produce the required behavioral improvements, it seems that in general, it should be possible in theory for shifts to provide the required benefits. For example, if the focal cue is on stimuli in the upper right quadrant, if all receptive fields in the model covering the other quadrants were fully shifted to the upper right quadrant, wouldn't the readout from the model obviously improve?

With regards to the linear classifiers implemented by the authors, it seems there are many important details that need to get spelled out. Things like how exactly was the final predicted category obtained? Was a 20-way classification scheme used, or some winner-take-all scheme from 20 separate binary classifiers used? Was there any regularization used to fit the weights of the logistic regression? If not, how can we be confident that the amount of training data was sufficient? Details on the classification methods are important for interpreting how upstream computational changes impact the behavioral output from the model.

2. Reframing and/or exploring other models.

In light of the uncertainties about how general the conclusions can be from the modeling results, the authors may wish to consider refocusing/reframing their work. Instead of attempting to make strong general conclusions about the relationship between behavior and computational mechanisms, the author might reframe their work into more of a computational and theory-building exercise.

In particular, the authors show interesting insights regarding how a Gaussian applied to an early layer of a CNN can have effects that propagate through the network and which manifest as complex, rich RF tuning changes elsewhere in the model (e.g. as the authors state, "RF shifts are a result of the signal gain…") (see also Compte Cerebral Cortex 2006). The distinction between architecture (internal computational mechanisms) and what might manifest at the level of input-output behavior of a unit is an important one, as the authors discuss (p. 17, lines 257-268). Certainly, a number of previous studies have adopted the latter mindset. Note that doing so is not necessarily inconsistent with the former "mechanistic" mindset (they are just different modeling stances one can take). Emphasizing and generating insights from adopting a "mechanistic" mindset is valuable research output.

Thus, one suggestion is to reframe the work to focus more on this insight and think about the theoretical ramifications (and/or concrete experimental predictions) that might stem from this. Related to this effort might be to consider other models. For example, it would be interesting to understand the similarities and differences between the 'attentional field' model (Reynolds, Heeger) and the current model. As another example, it would be interesting to consider models in which the "Gaussian gain" is applied at the end (deep layers) and effects propagate backwards, as opposed to being applied at the beginning and effects propagating forwards.

[Editors' note: further revisions were suggested prior to acceptance, as described below.]

Thank you for resubmitting your work entitled "Gain, not concomitant changes in spatial receptive field properties, improves task performance in a neural network attention model" for further consideration by *eLife*.

Reviewer #2 and I have re-evaluated your revised submission and we both agree that it is much improved and makes a strong contribution to the literature. There are a few remaining issues that need to be addressed that are captured in Reviewer #2's comments, and I'd ask you to address them before a final decision is made. Please let me know if you have any questions, and thanks.

*Reviewer #2 (Recommendations for the authors):*

I have reviewed the revisions and the authors' responses.

Overall, the revisions are solid and substantial. The additions to the paper are useful, in particular, the new analyses exploring exactly how spatial information is included, restricted, or "averaged over", which provide insights into the potential equivalency of the gain effects and the selection (or suppression) of irrelevant information. In addition, the new framing and discussion provide helpful clarity over the original manuscript and make for a more straightforward read. The computational modeling performed in this paper represents a novel and well-executed foray into a challenging area. While it does not necessarily definitively resolve the issues at stake, I believe it provides and demonstrates an interesting and non-trivial approach.

A few critical comments on the revisions:

The authors state, "At larger scales or in other tasks there are theoretical reasons to expect that task performance will improve due to these effects". In this sentence, the referent of "these effects" is a bit unclear. Also, I am not sure what 'larger scales' refers to. Finally, it would be useful to give specific examples of what types of 'other tasks' might be implied by the sentence here.

In response to the reviews, the authors added some detail on the linear classifier methods, but the text is still not sufficient. More detail on exactly what was implemented seems important. For example, details on the "under the curve" quantification. In addition, perhaps there is a misspecification of the methods: As the authors state, "Weights were fit using logistic regression with an L2 loss and no regularization, using scikit-learn and the LIBLINEAR package (Pedregosa et al., 2011)." However, if I understand correctly, for each category, there are effectively 512 parameters that need to get learned and only 100 instances to train these parameters. Hence, if no regularization were used, this presumably would lead to a very noisy (overfit) solution. Perhaps the authors meant that they used L2 regularization (and the default settings for the regularization hyperparameter in the scikit-learn package)? Note that logistic regression implies a probabilistic formulation of the classification problem, and would seem to suggest that the authors don't actually mean that they used an L2 loss (which implies additive Gaussian noise). Hopefully, these important analysis choices and details can be clarified and do not have larger ramifications for the rest of the paper.

---

## [Author Response]

Essential revisions:We were all very enthusiastic about the approach and the findings. During our discussions, we collectively focused on several issues regarding the generality of the conclusions and the methods used to arrive at them. Below is a list of the main issues:1) The Detection vs discrimination. Having at least one other task would allow you to determine if the gain story is specific to detection or if the type of modulation depends on the task. In either case, results from another task would be interesting and would greatly improve the generality and theoretical impact of the claims.

We agree that generalization to a discrimination task is worth testing. To ensure our results generalize better we ran our analysis against such a task using an analog to a two-interval forced choice task and found qualitatively similar results to the detection task [Figure 8 Supplement 2].

2) Justifying the spatial invariance of the readout mechanism and potentially testing other schemes.

Agreed that the assumption of spatial invariance in the original model is important to test. To ensure that our results were not affected by the choice to read out from a spatially collapsed final layer, we now report two new analyses: first, the result for a model trained to read out from the full spatial output (7 x 7 x 512) [292-308] and second, the result for a model trained to read out only from the target quadrant while masking out the other four. The results from these analyses support our initial conclusions, please see the text for the rationale and results [Figure 8 Supplement 1].

3) What happens when gain isn't just injected into "V1", but into other layers? For example, Lindsay has shown that this matters in the context of CNNs performing attention tasks, and the rationale for injecting a change so early in the system isn't entirely clear.

We’ve clarified in the text that the propagated gain test which is applied at the last layer of the CNN (Figure 8) demonstrates that for this particular model the choice of layer only changes whether the receptive field properties are changed, not the task performance [274-283]. We’ve also added a more complete discussion of the rationale for injecting gain into the earlier layers [442-448] and the Lindsay paper specifically [453-456].

4) Others, like Compte and Wang 2006, have also explored this question using other architectures. Given that the CORnet-z model does not include recurrent dynamics within each layer, it will be important to discuss and consider the implications of the chosen network and implications for the generality of the conclusions.

We’ve added a more complete discussion of this issue in the discussion [411-424].

Reviewer #1 (Recommendations for the authors):1) The model requires a more detailed presentation in the text (now lines 93,94). A succinct description of how it is built should be provided so the reader does not have to look up the references for a simple, broad-picture understanding. It is important to mention that the model has been built and trained previously based on the same set of images that will be used here, and the weights are available, so the only further training done here is the linear classifier on the model outputs to detect image categories upon presentation of "composite" grids. It took a while for me to understand this and to see the difference between base images and composite images. There is also a confusing thing about the neural model in Figure 2d: it pictures the ReLU occurring after the max pooling step, but in Figure 1 of Kubilius et al. (2018) it occurs before the max pooling step. The sentence ("Unit activations were measured after the convolution, prior to the max pooling step") in the caption of Figure 2 is also confusing: how is unit activity being measured before the ReLU non-linearity that transforms inputs into output rates?

We’ve added a better description of the model in both the methods [564-590] and results [109-113]. We thank the reviewer for pointing out the error in the figure and text concerning the order of the relu and maxpool operations and have fixed these.

2) The manuscript emphasizes 3 different RF modulations: scaling, shift, and shrinkage, but then only two of them are specifically isolated with network simulations. RF shrinkage is not addressed. This is not really a problem, but it would be nice to have full "symmetry" in the manuscript. Is there a specific reason why this modulation can not be assessed to make it more parallel to the rest?

We very much appreciate the suggestion to make the paper symmetrical and have added a new figure and series of analyses showing the effects of shrinkage [Figure 6, 222-245].

3) References are required in lines 145 ("…behavioral benefits of attention") and 250 ("…as others have suggested").

Citations have been added for each statement.

4) In general, scatter plots in the figures should indicate what are the individual data points presented. I think in some cases are units, in other cases they are categories. This should be clearly indicated.

We’ve clarified whether data points indicate units or categories for all figures.

Reviewer #2 (Recommendations for the authors):1. On the issue of the readout.As mentioned in the public review, it appears that the conclusions are heavily dependent on the choice of readout mechanism. Thus, it seems a bit premature at this point to make strong general conclusions about which tuning properties are critical for behavioral improvements. Many recent studies have shown spatial tuning (i.e. large but limited RF sizes) in the high-level visual cortex; hence, in the neural network observer model, it is not clear why the readout (linear classification) is performed on unit activity that has been summed fully across visual space.

We agree that it is important to consider models with some spatial tuning at the decision stage to more accurately model ventral temporal cortical inputs that are expected to have such tuning. We have added a model variant to address this issue and have arrived at similar conclusions, see [292-301] for details.

Now, it is an open interesting empirical question whether perceptual decisions from the visual system are based on fully spatially invariant units. But note that if that were the case, isn't one counterintuitive implication that our perceptual decisions can be corrupted by visual stimuli far away from the relevant decision zone?

Yes, we agree. We would clarify though that we are considering both visual processing and the decision stage [see 424-440]. At the decision stage, uncertainty over which visual signal to base a decision on is a problem that attention has been proposed to solve (e.g. Pelli, 1985). In this context, set size effects in which increasing the size of stimulus arrays across the visual field increases response times (even for covert search), can be viewed as the effect of uncertainty introduced by visual stimuli far from a search target. Or, for example in our own work, in which cueing one of four spatially separate targets improves contrast discrimination thresholds compared to distributing attention (Pestilli et al., 2011), even though distractors are far from the post-cued decision target.

If we entertain the possibility that the readout can be from units that have spatial specificity, then it remains to be seen what sort of modeling outcomes there might be.

Please see [309-319] where we show that the modeling outcomes for considering spatially specific responses does not qualitatively change our conclusions.

Even though the authors show, in the context of their particular model, that tuning shifts do not produce the required behavioral improvements, it seems that in general, it should be possible in theory for shifts to provide the required benefits. For example, if the focal cue is on stimuli in the upper right quadrant, if all receptive fields in the model covering the other quadrants were fully shifted to the upper right quadrant, wouldn't the readout from the model obviously improve?

We do agree that, in principle, tuning shifts could provide required benefits. We would like to clarify that we are considering the tuning shifts that occur because of downstream effects of increasing gain of a magnitude required to match human performance benefits with attention cueing. In our reframing of the manuscript (see below), we have substantially rewritten the text to clarify this particular point, and that our conclusion is *not* that tuning shifts of any magnitude won’t have behavioral effects [394-395 and 441-467], but is specific to those that occur with gain changes. We have also specifically tested the case pointed out above in which only receptive fields covering the target are considered and receptive field shifts of the magnitude found for the gain change are considered, We find that the magnitude of these changes is not enough to induce task performance benefits for the observer model. See [320-328].

With regards to the linear classifiers implemented by the authors, it seems there are many important details that need to get spelled out. Things like how exactly was the final predicted category obtained? Was a 20-way classification scheme used, or some winner-take-all scheme from 20 separate binary classifiers used? Was there any regularization used to fit the weights of the logistic regression? If not, how can we be confident that the amount of training data was sufficient? Details on the classification methods are important for interpreting how upstream computational changes impact the behavioral output from the model.

These details have now been added in the results [117-119] and methods [570-590]. We believe that the training data was sufficient because the validation set showed stable classification accuracy [118-119].

2. Reframing and/or exploring other models.In light of the uncertainties about how general the conclusions can be from the modeling results, the authors may wish to consider refocusing/reframing their work. Instead of attempting to make strong general conclusions about the relationship between behavior and computational mechanisms, the author might reframe their work into more of a computational and theory-building exercise.In particular, the authors show interesting insights regarding how a Gaussian applied to an early layer of a CNN can have effects that propagate through the network and which manifest as complex, rich RF tuning changes elsewhere in the model (e.g. as the authors state, "RF shifts are a result of the signal gain…") (see also Compte Cerebral Cortex 2006). The distinction between architecture (internal computational mechanisms) and what might manifest at the level of input-output behavior of a unit is an important one, as the authors discuss (p. 17, lines 257-268). Certainly, a number of previous studies have adopted the latter mindset. Note that doing so is not necessarily inconsistent with the former "mechanistic" mindset (they are just different modeling stances one can take). Emphasizing and generating insights from adopting a "mechanistic" mindset is valuable research output.Thus, one suggestion is to reframe the work to focus more on this insight and think about the theoretical ramifications (and/or concrete experimental predictions) that might stem from this. Related to this effort might be to consider other models. For example, it would be interesting to understand the similarities and differences between the 'attentional field' model (Reynolds, Heeger) and the current model. As another example, it would be interesting to consider models in which the "Gaussian gain" is applied at the end (deep layers) and effects propagate backwards, as opposed to being applied at the beginning and effects propagating forwards.

Thank you for this helpful suggestion. We have substantially modified the manuscript throughout to reframe as suggested. In particular, please note the new title, abstract, changes in the introduction [36-38, 49-52, 59-64] and re-written discussion. We also have extended our discussion of the alternate model (what we call propagated gain model) of adding gain at the last stages (end, deep layers) of the CNN [274-328]. The RH models consider the effects of gain in a similar fashion but only on a single layer model, which we now mention in the discussion [435-437]

[Editors' note: further revisions were suggested prior to acceptance, as described below.]

Reviewer #2 (Recommendations for the authors):I have reviewed the revisions and the authors' responses.Overall, the revisions are solid and substantial. The additions to the paper are useful, in particular, the new analyses exploring exactly how spatial information is included, restricted, or "averaged over", which provide insights into the potential equivalency of the gain effects and the selection (or suppression) of irrelevant information. In addition, the new framing and discussion provide helpful clarity over the original manuscript and make for a more straightforward read. The computational modeling performed in this paper represents a novel and well-executed foray into a challenging area. While it does not necessarily definitively resolve the issues at stake, I believe it provides and demonstrates an interesting and non-trivial approach.A few critical comments on the revisions:The authors state, "At larger scales or in other tasks there are theoretical reasons to expect that task performance will improve due to these effects". In this sentence, the referent of "these effects" is a bit unclear. Also, I am not sure what 'larger scales' refers to. Finally, it would be useful to give specific examples of what types of 'other tasks' might be implied by the sentence here.

We agree that these additional details help clarify this important discussion point. We have clarified the meaning of each term and specifically mention judgments of spatial position as an example of a task that should theoretically show a larger benefit from shift/shrinkage of receptive fields. [new text at lines 460-472].

In response to the reviews, the authors added some detail on the linear classifier methods, but the text is still not sufficient. More detail on exactly what was implemented seems important. For example, details on the "under the curve" quantification.

We have added additional detail to this section, explaining how the AUC is calculated and how it can be interpreted [lines 681-690].

In addition, perhaps there is a misspecification of the methods: As the authors state, "Weights were fit using logistic regression with an L2 loss and no regularization, using scikit-learn and the LIBLINEAR package (Pedregosa et al., 2011)." However, if I understand correctly, for each category, there are effectively 512 parameters that need to get learned and only 100 instances to train these parameters. Hence, if no regularization were used, this presumably would lead to a very noisy (overfit) solution. Perhaps the authors meant that they used L2 regularization (and the default settings for the regularization hyperparameter in the scikit-learn package)? Note that logistic regression implies a probabilistic formulation of the classification problem, and would seem to suggest that the authors don't actually mean that they used an L2 loss (which implies additive Gaussian noise). Hopefully, these important analysis choices and details can be clarified and do not have larger ramifications for the rest of the paper.

We thank the reviewer for raising these important concerns, which were partly due to a typo. We have fixed the typo to clarify that L2 regularization was used in training the output layers, specifically to reduce the risk of overfitting to the training data [Methods lines 673-680]. We have also made it more clear that the output layers were evaluated on a held-out validation set (median AUC 0.9), providing further evidence that the models are not overfit [Results lines 194-195] and we have copied this information into the methods to make it easier to find [Methods lines 679-680]